

# Land-surface forcing and anthropogenic heat modulate ozone by
# meteorology: A perspective from the Yangtze River Delta region
Chenchao Zhan [a], Min Xie [a,*]
[a] School of Atmospheric Sciences, CMA-NJU Joint Laboratory for Climate Prediction Studies,
Jiangsu Collaborative Innovation Center for Climate Change, Joint Center for Atmospheric Radar
Research of CMA/NJU, Nanjing University, Nanjing 210023, China

7  -------------------------------------------------------------------

[*] Corresponding author. minxie@nju.edu.cn (M. Xie)
**Abstract:** With the rapid advance in urbanization, land-surface forcing related to the urban
expansion and anthropogenic heat (AH) release from human activities significantly affect the urban
climate and in turn the air quality. Focusing on the Yangtze River Delta (YRD) region, a highly
urbanized place with sever ozone ($O_3$) pollution and complex geography, we estimate the impacts
of land-surface forcing and AH on meteorology (meteorological factors and local circulations) and
$O_3$ using the WRF-chem model, which can enhance our understanding about the formation of $O_3$
pollution in those rapidly developing regions with unique geographical features as most of our
results can be supported by previous studies conducted in other regions in the world. Regional $O_3$
pollution episodes occur frequently (26 times per year) in the YRD in recent years. These $O_3$
pollution episodes are usually under calm conditions characterized by high temperature (over 20 ℃),
low relative humidity (less than 80%), light wind (less than 3 m s$^{-1}$) and shallow cloud cover (less
than 5). In this case, high $O_3$ mainly appears during the daytime influenced by the local circulations
(the sea and the lake breezes). The change in land-surface forcing can cause an increase in 2-m
temperature ($T_2$) by maximum 3 ℃, an increase in planetary boundary layer height (PBLH) by
maximum 500 m and a decrease in 10-m wind speed ($WS_{10}$) by maximum 1.5 m s$^{-1}$, and surface $O_3$
can increase by maximum 20 μg m$^{-3}$ eventually. Furthermore, the expansion of coastal cities
enhances the sea-breeze below 500 m. During the advance of the sea-breeze front inland, the upward
air flow induced by the front makes well vertical mixing of $O_3$. However, once the sea-breeze is
fully formed, further progression inland is stalled, thus the $O_3$ removal by the low sea-breeze will
be weakened and surface $O_3$ can be 10 μg m$^{-3}$ higher in the case with cities than no-cities. The



expansion of lakeside cities can extend the lifetime of the lake-breeze from the noon to the afternoon.
Since the net effect of the lake-breeze is to accelerate the vertical mixing in the boundary layer, the
surface $O_3$ can increase as much as 30 μg m$^{-3}$ in lakeside cities. Compared with the effects from
land-surface forcing, the impacts of AH are relatively small. And the changes mainly appear in and
around cities where AH emission is large. There are increases in $T_2$, PBLH, $WS_{10}$ and surface $O_3$
when AH are taken into account, with the increment about 0.2 ℃, 75 m, 0.3 m s$^{-1}$ and 4 μg m$^{-3}$,
respectively. Additionally, AH can affect the urban-breeze circulations, meteorological factors and
$O_3$ concentration, but its effect on local circulations, such as the sea and the lake breezes, seems to
be limited.
**Key Words:** ozone; meteorology; local circulations; land-surface forcing; anthropogenic heat; the
Yangtze River Delta;

**1 Introduction**
Ozone ($O_3$) is a key constituent in the atmosphere, and is deeply relevant to climate (Worden
et al., 2008), biosphere (Van Dingenen et al., 2009) and human health (Jerrett et al., 2009). $O_3$ acts
quite differently in different parts of the atmosphere, often described as being "good up high and
bad nearby". $O_3$ in the stratosphere helps protect life on earth from strong ultraviolet radiation.
However, high $O_3$ in the troposphere is harmful to human respiratory system and the growth of
vegetation, and thereby the tropospheric $O_3$ has long been regarded as an important air pollutant
(Young et al., 2013).
Tropospheric $O_3$ is a secondary air pollutant, which is formed by a series of complex chemical
reactions (Chameides and Walker, 1973; Xie et al., 2014) of precursor gases such as nitrogen oxides
($NO_x$=NO+$NO_2$) and volatile organic compounds (VOCs) in combination with sunlight. The global
average lifetime of tropospheric $O_3$ is 20 to 25 days, and it will be reduced to 5 days in boundary
layer (Young et al., 2013). The relatively long lifetime of tropospheric $O_3$ favors regional/long-range
transport, and brings huge challenges to its control (Shao et al., 2006). $O_3$ levels considerably depend
on the variations in weather conditions because weather conditions play an important role in
determining the chemistry, dispersion and removal of $O_3$ (Jacob and Winner, 2009). Generally,
elevated $O_3$ occurs under warm dry weather with strong sunlight, high temperature, low relative
humidity and light wind speed (Zhang et al., 2015). Furthermore, weather conditions can have many





similarities in certain weather pattern (Buchholz et al., 2010; Zhan et al., 2019), and the main
weather patterns associated with $O_3$ episodes in China are tropical cyclones and continental
anticyclones (Wang et al., 2017).

$O_3$ levels as well as weather conditions in urban areas are of great concern simply because

urban areas have huge populations. A report from the United Nations pointed out that 69.6% of the
world's population will live in cities by 2050. The urbanization process has further increased urban
environmental hazards (Zhang et al., 2011), particularly in the most rapidly developing countries
like China (Liu and Tian, 2010). Because of historical and cultural factors, many cities have similar
topography, usually along the coast, close to mountains or in basins. For these cities, the local
circulations induced by thermal contrast of the topography, such as sea-land breezes, mountain-
valley breezes and lake-land breezes, will have an important impact on air quality of the city,
especially when the dominant background weather system is weak (Crosman and Horel, 2010).
Examples can be found around the world. Ding et al. (2004) simulated the main features of the sea-
land breezes during a multiday episode in the Pearl River Delta (PRD) region, and found that the
sea-land breezes play a crucial role in transporting air pollutant between inland and coastal cities.
Miao et al. (2015) studied the effects of mountain-valley breezes on boundary layer structure in the
Beijing-Tianjin-Hebei (BTH) region, suggesting that the mountain-valley breezes are vital to the
vertical transport and distribution of air pollutants in Beijing. Wentworth et al. (2015) identified a
causal link between lake-breeze and $O_3$ in the Greater Toronto Area that the daytime $O_3$ maxima
was 13.6-14.8 ppb higher on lake breeze days than no-lake breeze days.

The land-surface forcing and anthropogenic heat (AH) of a city also affect the atmospheric

state and compositions above it (Yu et al., 2012; Oke et al., 2017). The land-surface forcing changes
chiefly come from the urban expansion (typically from vegetation to impervious surface), which
directly changes the surface physical properties (e.g., albedo, surface moisture and roughness) and
thereby significantly affects the meteorology and in turn the air quality. Li et al. (2019) found that
increases in thermal inertia, surface roughness and evapotranspiration due to urban expansion can
lead to an increase in $O_3$ by up to 5.6 ppb in Southern California. AH is an important waste by-
product of urban metabolism. Nearly all energy consumed by human activities will be dissipated as
heat within Earth's land-atmosphere system (Flanner, 2009; Sailor, 2011) that is then "injected" into
the    energy    balance    processes.    Ryu    et    al.    (2013a)    reported    that    AH    affects    the



characteristics/structures of boundary layer and local circulations, resulting in an increase of $O_3$ by
3.8 ppb in the Seoul metropolitan area.
These previous studies separately investigated the impact of local circulations, land-surface
forcing and AH on meteorology and air quality, usually focusing on a specific megacity. However,
local circulations, land-surface forcing and AH can work together in near-calm conditions. And the
role of multi-scale atmospheric circulations associated with the abovementioned factors in regional
meteorology and air quality of city clusters is unclear. Actually, complex interactions exist widely
among these thermally-driven circulations and the effects can even spread from one city to nearby
areas. For example, Zhu et al. (2015) demonstrated that the meteorological conditions and air quality
over Kunshan are significantly affected by Shanghai urban land surface forcing (Kunshan is located
downstream of Shanghai, with a straight-line distance of about 50 km). Therefore, assessing the
effects of land-surface forcing and AH (The topography rarely changes.) in the city cluster is
meaningful, which helps understand the connection between urban development, local meteorology
and regional air quality.
The Yangtze River Delta (YRD) region, located on the western coast of the Pacific Ocean
(Figure 1a), has undergone accelerated urbanization process and rapid economic development over
the past decades, and is now one of the largest economic zones in the world. It includes the areas of
the southern part of Jiangsu Province, the northern part of Zhejiang Province and the eastern part of
Anhui Province, with 26 mega/large cities such as Shanghai, Hangzhou and Nanjing (Figure 1b).
With dense population and huge energy consumption, this area is now suffering from air quality
deterioration (Ding et al., 2013; Xie et al., 2017), especially severe $O_3$ pollution in recent years
(Zhan et al., 2020, 2021). It was reported that 16 out of the 26 typical cities in the YRD failed to
meet the urban national standard for $O_3$ in 2017 (Bulletin on the state of China's ecological
environment in 2018, http://www.cleanairchina.org/ product/9943.html), and to make matters worse,
$O_3$ concentration has been rising in this region during the past few years (Li et al., 2020; Wang et
al., 2020). The YRD region is deeply affected by the East Asian monsoon, and has complex weather
like other mid-latitude regions in the world. Sever air pollution and unique geography make this
area an ideal place for studying the complex interactions between the atmosphere and human
activities.
In this study, the impacts of land-surface forcing and AH on meteorology in the central YRD





region, and how these impacts further modulate O$_3$ are investigated using the Weather Research and
Forecasting model coupled to Chemistry (WRF-Chem). These results fill the knowledge gap about
the formation of O$_3$ pollution in this region and provide valuable insight for other rapidly developing
regions with complex geography in the world. The remainder of this paper is organized as follows.
Sect. 2 gives a detailed description about the observation data, the model setup and experimental
design. The main results, including the characteristics of O$_3$ pollution episodes, the model evaluation
and the response of O$_3$ to land-surface forcing and AH, are presented in Sect. 3. Summary and
conclusions are given in Sect. 4.

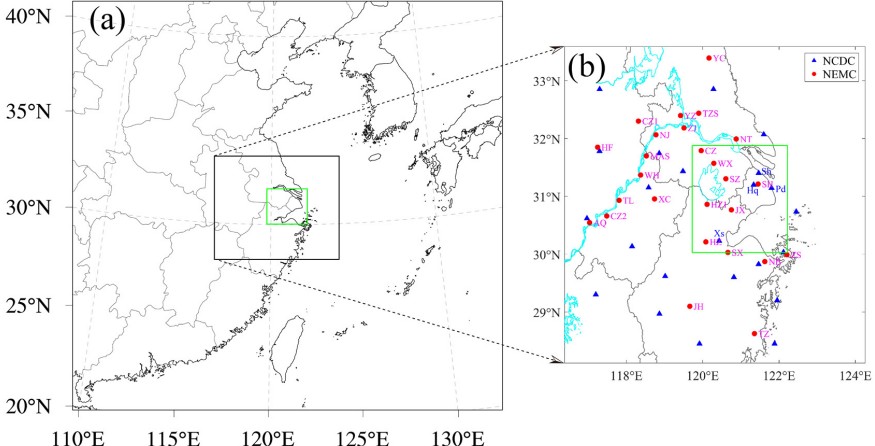


**Figure 1.** (a) Three nested WRF-Chem domains, (b) the locations of 26 cities (red dots) and weather
stations (blue triangles) in the YRD. The green rectangular regions represent the innermost domain
and also the central YRD region. These cities in (b) include: the megacity Shanghai (SH); Hangzhou
(HZ), Ningbo (NB), Jiaxing (JX), Huzhou (HZ1), Shaoxing (SX), Jinhua (JH), Zhoushan (ZS) and
Taizhou (TZ) located in Zhejiang Province; Nanjing (NJ), Wuxi (WX), Changzhou (CZ), Suzhou
(SZ), Nantong (NT), Yancheng (YC), Yangzhou (YZ), Zhenjiang (ZJ) and Taizhoushi (TZS) located
in Jiangsu Province; and Hefei (HF), Wuhu (WH), Maanshan (MAS), Tongling (TL), Anqing (AQ),
Chuzhou (CZ1), Chizhou (CZ2) and Xuancheng (XC) located in Anhui Province.

**2 Materials and methods**



## 2.1 Surface observations

**2.1 Surface observations**
Hourly $O_3$ concentrations monitored by the National Environmental Monitoring Center
(NEMC) of China are used in this study. These data strictly follow the national monitoring standards
HJ 654-2013 and HJ 193-2013 (http://www.cnemc.cn/jcgf/dqhj/), and can be available at
https://quotsoft.net/air/, a mirror of data from the official NEMC real-time publishing platform
(http://106.37.208.233:20035/). The nationwide observation network initially operated in 74 major
cities in 2013, and it has grown to more than 1,500 stations covering 454 cities by 2017 (Lu et al.,
2018). The urban hourly $O_3$ concentrations are average results of measurements at all monitoring
sites for each city. The maximum daily 8-h running average (MDA8) $O_3$ concentrations are then
calculated based on the hourly $O_3$ concentration with more than 18-h measurements in the day (Liao
et al., 2017).
Meteorological data are provided by the National Climatic Data Center (NCDC), including
temperature, wind speed and direction, and relative humidity, etc. These data as well as the technical
documents recording the quality control, data collection and archive can be available at
ftp://ftp.ncdc.noaa.gov/pub/data/noaa/isd-lite/. Locations of surface observation stations are shown
in Figure 1b. Specifically, the meteorological stations in the innermost domain include Pudong (Pd),
Shanghai (Sh), Hongqiao (Hq) and Xiaoshan (Xs).

## 2.2 MODIS-based and USGS land use classifications

**2.2 MODIS-based and USGS land use classifications**
To investigate the impact of land-surface forcing on regional meteorology and $O_3$ evolution in
the YRD, the two land use categories defaulted in WRF (MODIS-based and USGS land use
classifications) are used to set up the first two sensitivity simulations (Table 2). The MODIS-based
land cover product was created from 500-m MODIS Terra and Aqua satellite imagery (Friedl et al.,
2010), and replaced USGS as the default settings in WRF since version 3.8. The USGS data
primarily derived from the Advanced Very High Resolution Radiometer (AVHRR) from 1992 to
1993 at 1-km spatial resolution (Loveland et al., 2000), which is much earlier than the MODIS data.
Figure 2 presents the land cover maps in the innermost domain. Apparently, urban fraction with
MODIS is much higher than USGS, indicating rapid urbanization in recent decades in the YRD.
The differences in urban land-surface forcing between USGS and MODIS mainly depend on urban
expansion. Additionally, the Finer Resolution Observation and Monitoring-Global Land Cover in
2015 (From-GLC_2015), which can be considered as one of the latest (2015) and finest (30-m) land


cover datasets (Gong et al., 2019), is quite consistent with the performance of MODIS in this region.
This further confirms that urban fraction with MODIS is close to the reality.

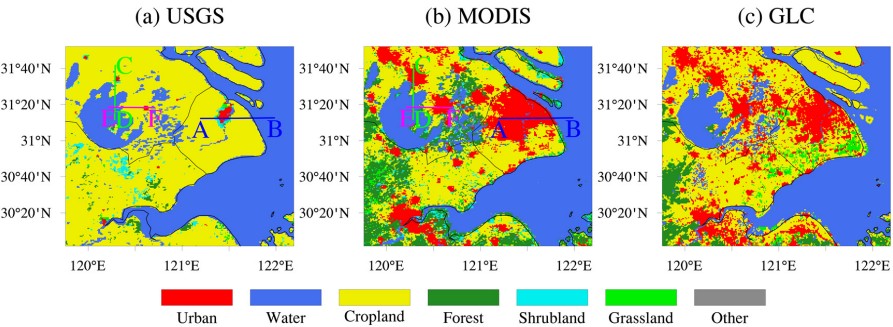

**Figure 2.** Land cover maps in the innermost domain, including the result of (a) USGS, (b) MODIS,
and (c) From-GLC_2015.

**2.3 Anthropogenic heat flux modeling**

Another simulation involved the urban canopy model with the gridded AH fluxes is conducted

to estimate AH release in the central YRD. The AH fluxes are mainly the result of chemical energy
or electrical energy that are converted to heat, thereby they can be quantified using the top-down
energy inventory method. Based on the statistics data of energy consumption in 2016, the AH fluxes
were calculated, and then were gridded as 144 rows and 144 columns with a resolution at 2.5 arcmin
using population density in China. Details on the calculation as well as the distribution of AH fluxes,
and how to add AH fluxes into the urban canopy can refer to Xie et al. (2016a, b). Figure 3 gives
the spatial distribution of AH fluxes in the innermost domain. In the urban areas, the AH fluxes
usually exceed 20 W m$^{-2}$. Some big cities, like Shanghai, can have a value of AH flux as high as
W m$^{-2}$. Except for the urban areas, the AH fluxes are generally less than 5 W m$^{-2}$ in most parts
of the YRD region. In particular, in those places where there is no human activity, the AH flux is 0.





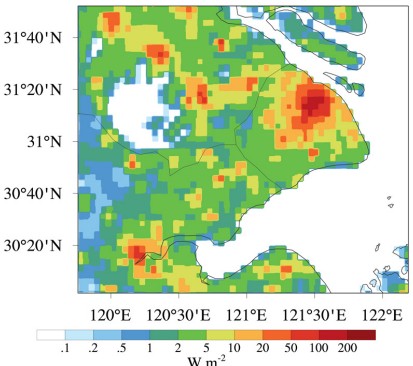


**Figure 3.** Spatial distribution of anthropogenic heat fluxes in the innermost domain.

**2.4 Model set-up and experimental designs**
The WRF-Chem model is a fully coupled online numerical weather prediction model with
chemistry component (Grell et al., 2005), in which air quality and the meteorological component
use the same coordinates, transport schemes and physics schemes in space and time. In this study,
the WRF-Chem version 3.9.1 is applied. The initial and boundary conditions of meteorological
fields are from the National Centers for Environmental Prediction (NCEP) global final analysis
fields every 6 h with a spatial resolution of $1° × 1°$. There are 32 vertical levels extending from the
surface to 100 hPa with 12 levels located below 2 km to resolve the boundary layer processes.
Furthermore, the domain and options for physical and chemical parameterization schemes are
summarized in Table 1. The anthropogenic emissions are provided by the Multiresolution Emission
Inventory for China (MEIC) in 2017 with a resolution of 0.25° (http://meicmodel.org/), which
includes 10 air pollutants and $CO_2$ from power, industry, residential, transportation and agriculture
sectors. The biogenic emissions are estimated online by the Model of Emissions of Gases and
Aerosols from Nature (MEGAN) in WRF-Chem (Guenther et al., 2006).

**Table 1.** The domains and major options for WRF-Chem

| Items | Contents |
| --- | --- |
| Dimensions (x, y) | (101, 96), (146, 121), (236, 206) |
| Grid spacing (km) | 25, 5, 1 |



| Time step (s) | 75 |
|---|---|
| Microphysics | Purdue Lin microphysics scheme (Chen and Sun, 2002) |
| Longwave radiation | RRTM scheme (Mlawer et al., 1997) |
| Shortwave radiation | Goddard scheme (Kim and Wang, 2011) |
| Surface layer | Revised MM5 Monin-Obukhov scheme |
| Land-surface layer | Noah land-surface model (Chen and Dudhia, 2001) |
| Planetary boundary layer | YSU scheme (Hong et al., 2006) |
| Cumulus parameterization | Grell 3D ensemble scheme (Grell and Devenyi, 2002) |
| Gas-phase chemistry | RADM2 (Stockwelll et al., 1990) |
| Photolysis scheme | Fast-J photolysis (Fast et al., 2006) |
| Aerosol module | MADE/SORGAM (Schell et al., 2001) |


As shown in Table 2, three numerical experiments are performed to study the effects of land-
surface forcing and AH on meteorology and $O_3$ in the YRD. The MODIS_noAH experiment is a
control simulation with commonly used settings. Compared with MODIS_noAH, USGS_noAH
selects the USGS data at run-time through the geogrid program. Thus, the difference between the
modeling results of MODIS_noAH and USGS_noAH can illustrate the changes caused by land
cover. As for the impact of AH, it can be identified by comparing the modeling results of
MODIS_withAH and MODIS_noAH. All three simulations run from 00:00 on 21 May to 00:00 on
4 June in 2017 with the first 88 h as spin-up time, using the same physical and chemical
parameterization schemes (Table 1).

**Table 2.** The three numerical experiments.

| Cases | Land use categories | Whether to add AH |
|---|---|---|
| MODIS_noAH | MODIS-based | No |
| USGS_noAH | USGS | No |
| MODIS_withAH | MODIS-based | Yes |


**2.5 Model evaluation**



The simulation results in the innermost domain, including $O_3$ concentration, 2-m air
temperature ($T_2$), relative humidity (RH), 10-m wind speed ($WS_{10}$) and 10-m wind direction ($WD_{10}$)
are examined against the surface observations described in Sect. 2.1. The statistical metrics,
including the mean bias (MB), root mean square error (RMSE) and correlation coefficient (COR),
are used to evaluate the model performance. They are defined as follows:

$$MB = \frac{1}{N}\sum_{i=1}^{N}(S_i - O_i),$$

(1)

$$RMSE = \sqrt{\frac{1}{N}\sum_{i=1}^{N}(S_i - O_i)^2},$$

(2)

$$COR = \frac{\sum_{i=1}^{N}(S_i - \bar{S})(O_i - \bar{O})}{\sqrt{\sum_{i=1}^{N}(S_i - \bar{S})^2}\sqrt{\sum_{i=1}^{N}(O_i - \bar{O})^2}},$$

(3)

where $S_i$ and $O_i$ are the simulations and observations, respectively. N is the total amount of valid
data, and $\bar{S}$ and $\bar{O}$ represent the average of simulations and observations, respectively. Generally,
the model performance is acceptable if the values of MB and RMSE are close to 0, and that of COR
is close to 1 (Xie et al., 2016a, b; Zhan et al., 2020).

**3 Results and discussions**
**3.1 Regional $O_3$ pollution episodes in the YRD**

Under adverse weather conditions, $O_3$ pollution episodes occur frequently in the YRD (Gao et

al., 2020; Zhan et al., 2021). Sometimes, $O_3$ pollution can spread throughout the YRD and cause
regional $O_3$ pollution, affecting an area of up to 3.5 million square kilometers and harming more
than 200 million people. Based on the surface $O_3$ observations, we define the regional $O_3$ pollution
in the YRD as when more than half of the 26 typical cities in the YRD fail to meet the national $O_3$
standard (In China, the national ambient air quality standard for MDA8 $O_3$ is 160 µg m$^{-3}$), and then
sort out all regional $O_3$ pollution episodes and the corresponding weather patterns from 2015 to
2019 (Table S1). There were 20, 19, 34, 28 and 30 regional $O_3$ pollution cases in the YRD from
2015 to 2019, respectively. These cases mainly occurred in April to October of each year, and were
usually related to high pressure, uniform pressure field and typhoon activity.



Figure 4 further displays the monthly distribution of meteorological factors during the day
(from 8:00 to 20:00 local time) when regional $O_3$ pollution occurs in the YRD. All the variables
show significant monthly variations. The highest (lowest) temperature is found in July (April), and
the relative humidity is highest in June. This may be related to the Meiyu in June, and the hot weather
in July as the YRD is usually dominated by the western Pacific subtropical high after Meiyu. As for
the cloud cover, the sky is covered with fewer clouds in October than other months. In addition,
southeast wind prevails in the YRD from April to October under the influence of monsoon climate.
As shown in Figure 4, $O_3$ pollution episodes are likely to occur in the YRD on days when the
temperature exceeds 20 ℃ (Figure 4b), the relative humidity is less than 80% (Figure 4c), the cloud
cover is less than 5 (Figure 4d), and the wind speed is less than 3 m s$^{-1}$ (Figure 4e). Interestingly,
the local circulations induced by thermal differentiation is clearest when in absence of clouds,
radiative heating is strongest and wind is weakest. Thus, both $O_3$ pollution and local circulations
tend to appear in calm conditions characterized by high temperature, cloudless sky and weak wind,
and the local circulation will inevitably have an impact on the distribution of $O_3$ in this case.



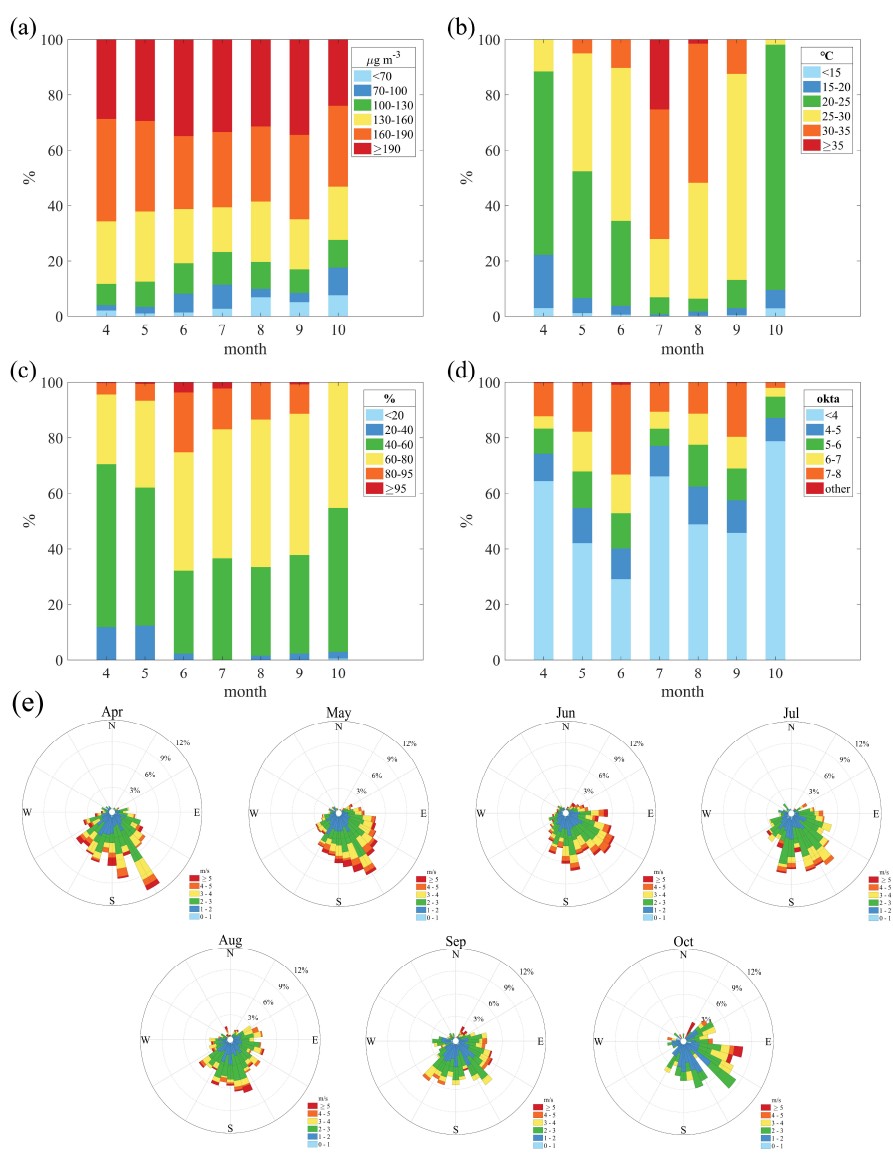


**Figure 4.** The monthly distribution of (a) O₃, (b) temperature, (c) relative humidity, (d) cloud cover,
and (e) wind speed and direction during the daytime (8:00 to 20:00 LT) when regional O₃ pollution
occurs in the YRD.

**3.2 Case selection**
**3.2.1 Case for O₃ pollution episode**



For simplicity but without loss of generality, the longest-lasting regional $O_3$ pollution case in
Table S1 is selected to investigate the impacts of land-surface forcing and AH on meteorology and
$O_3$ pollution in the YRD. This 10-day regional $O_3$ pollution episode occurred from 25 May to 3 June
in 2017. During this period, an average of 18 out of the 26 cities experienced $O_3$ pollution every day,
and the MDA8 $O_3$ concentrations ranged from 168.1 to 205.1 µg m$^{-3}$. Moreover, the daily maximum
air temperature ranged from 28.5 to 33.9 ℃ over the central YRD (the innermost domain) under
high pressure/uniform pressure field (Figure S1). This case meets the requirements of calm weather
and high $O_3$ concentration. And the relatively long duration also provide a representative result.
**3.2.2 Evaluation of model performance**
In this study, three numerical experiments are conducted using WRF-Chem (Sect. 2.4) during
the period of the previously mentioned $O_3$ episode. The simulation results are validated in the
innermost domain by comparing with the observational data. Table 5 presents the statistical metrics
in meteorological variables that includes 2-m air temperature ($T_2$), relative humidity (RH), 10-m
wind speed ($WS_{10}$) and direction ($WD_{10}$). Figure 5 further illustrates time series comparisons
between these meteorological factors and their modeling results. $T_2$ is reasonably well simulated as
the mean CORs (the mean of all the sites) are 0.875, 0.865 and 0.863 in MODIS_noAH,
USGS_noAH and MODIS_AH, respectively. The small negative MBs at all sites suggest that our
simulations underestimate $T_2$ to some extent, though this light underestimation is acceptable because
of the small mean RMSE (2.3, 3.1 and 2.3 ℃). The mean MBs for $T_2$ in USGS_noAH,
MODIS_noAH and MODIS_AH are -2.4, -1.0, and -0.8 ℃, indicting an improvement in
temperature when new land use and AH are taken into account. These results can be confirmed by
Figure 5a. With respect to RH, the mean CORs are 0.823, 0.753 and 0.825 for the three numerical
experiments, respectively. All three simulations can well capture the diurnal variation of RH, but
have different performance on different sites (Figure 5b). In USGS_noAH, RH is overestimated at
all sites, especially Pudong site, and the mean MB is 11.2%. While RH is only overestimated at the
two coastal sites (Pudong and Shanghai) but underestimated at other two sites (Hongqiao and
Xiaoshan) in MODIS_noAH and MODIS_AH. Moreover, USGS_noAH has the highest mean
RMSE of RH (16.3%), followed by MODIS_AH (12.4%) and MODIS_noAH (12.1%). As for $WS_{10}$,
the modeling values are slightly overestimated at all sites in all three simulations. The
overestimation of $WS_{10}$ may partly be attributed to the unresolved terrain features by the default



surface drag parameterization causing an overestimation of wind speed in particular at low values
(Jimenez and Dudhia, 2012). Specially, $WS_{10}$ in USGS_noAH is the most overestimated, followed
by MODIS_AH and MODIS_noAH, with the mean MBs are 1.2, 1.0 and 0.8 m s$^{-1}$, respectively.
Additionally, a high mean MB is found to correspond to a high mean RMSE (1.9, 1.8 and 1.7 m s$^{-}$
$^1$) in our simulations. In terms of $WD_{10}$, the model captures well the shift in wind direction during
the study period (Figure 5d). Thus, our modeling results of wind speed and direction basically reflect
the characteristics of wind fields. In summary, both the statistical metrics in Table 3 and time series
in Figure 5 illustrate that all the numerical experiments can reflect the major characteristics of
meteorological conditions during this $O_3$ pollution episode. Nevertheless, using new land-use data
and adding AH can reduce the underestimation of $T_2$ and the overestimation of RH and $WS_{10}$ to
some extent.





**Table 3.** Statistical metrics in meteorological variables between observations and simulations.

| Variables | Site | $\bar{O}^a$ | MODIS_noAH | | | | | USGS_noAH | | | | MODIS_AH | | | |
|---|---|---|---|---|---|---|---|---|---|---|---|---|---|---|---|
| | | | $\bar{S}^b$ | $MB^c$ | $RMSE^d$ | $COR^e$ | $\bar{S}$ | MB | RMSE | COR | $\bar{S}$ | MB | RMSE | COR |
| $T_2$ (°C) | Pd | 23.2 | 21.5 | -1.7 | 2.4 | 0.89 | 20.7 | -2.5 | 3.8 | 0.70 | 21.5 | -1.7 | 2.4 | 0.89 |
| | Sh | 24.6 | 23.9 | -0.7 | 2.2 | 0.87 | 22.5 | -2.1 | 2.7 | 0.90 | 24.2 | -0.5 | 2.3 | 0.84 |
| | Hq | 25.3 | 24.4 | -0.9 | 2.0 | 0.89 | 22.7 | -2.6 | 3.0 | 0.95 | 24.8 | -0.5 | 1.9 | 0.89 |
| | Xs | 25.9 | 25.1 | -0.8 | 2.4 | 0.85 | 23.8 | -2.2 | 2.8 | 0.91 | 25.5 | -0.4 | 2.4 | 0.83 |
| RH (%) | Pd | 69.1 | 77.7 | 8.6 | 13.5 | 0.81 | 86.2 | 17.2 | 23.4 | 0.45 | 77.7 | 8.7 | 13.3 | 0.83 |
| | Sh | 59.3 | 60.6 | 1.3 | 11.7 | 0.81 | 71.1 | 11.8 | 16.1 | 0.81 | 59.4 | 0.1 | 12.4 | 0.78 |
| | Hq | 59.5 | 57.7 | -1.8 | 9.8 | 0.88 | 70.6 | 11.1 | 14.5 | 0.89 | 56.2 | -3.3 | 9.8 | 0.89 |
| | Xs | 60.6 | 55.4 | -5.2 | 13.5 | 0.79 | 65.3 | 4.8 | 11.3 | 0.86 | 53.5 | -7.1 | 14.1 | 0.80 |
| $WS_{10}$ (m s$^{-1}$) | Pd | 4.1 | 4.1 | 0.0 | 1.4 | 0.47 | 5.5 | 1.3 | 2.1 | 0.35 | 4.2 | 0.1 | 1.3 | 0.51 |
| | Sh | 2.5 | 4.2 | 1.7 | 2.2 | 0.36 | 4.5 | 2.0 | 2.4 | 0.54 | 4.3 | 1.9 | 2.3 | 0.35 |
| | Hq | 3.7 | 3.9 | 0.2 | 1.2 | 0.54 | 3.9 | 0.2 | 1.2 | 0.53 | 4.2 | 0.5 | 1.3 | 0.50 |
| | Xs | 2.3 | 3.6 | 1.3 | 2.0 | 0.26 | 3.4 | 1.1 | 1.8 | 0.30 | 3.8 | 1.5 | 2.1 | 0.24 |
| $WD_{10}$ (°) | Pd | 160.4 | 136.1 | -26.2 | 78.7 | 0.42 | 148.1 | -14.3 | 55.1 | 0.72 | 137.3 | -24.7 | 77.5 | 0.42 |
| | Sh | 141.6 | 146.4 | 4.8 | 66.4 | 0.60 | 141.7 | 0.1 | 63.9 | 0.59 | 142.6 | 1.0 | 69.9 | 0.56 |
| | Hq | 159.7 | 140.2 | -23.4 | 80.2 | 0.46 | 153.1 | -10.6 | 74.9 | 0.52 | 142.8 | -20.4 | 91.8 | 0.29 |
| | Xs | 188.6 | 160.2 | -28.4 | 99.5 | 0.48 | 161.4 | -27.3 | 109.6 | 0.35 | 152.0 | -36.6 | 109.9 | 0.38 |

$^a$ $\bar{O}$ and $^b$ $\bar{S}$ indicate the average of observations and simulations, respectively. $^c$ MB indicates the mean bias, $^d$ RMSE indicates the root mean square error and $^e$ COR
indicate the correlation coefficient, with statistically significant at 99% confident level.






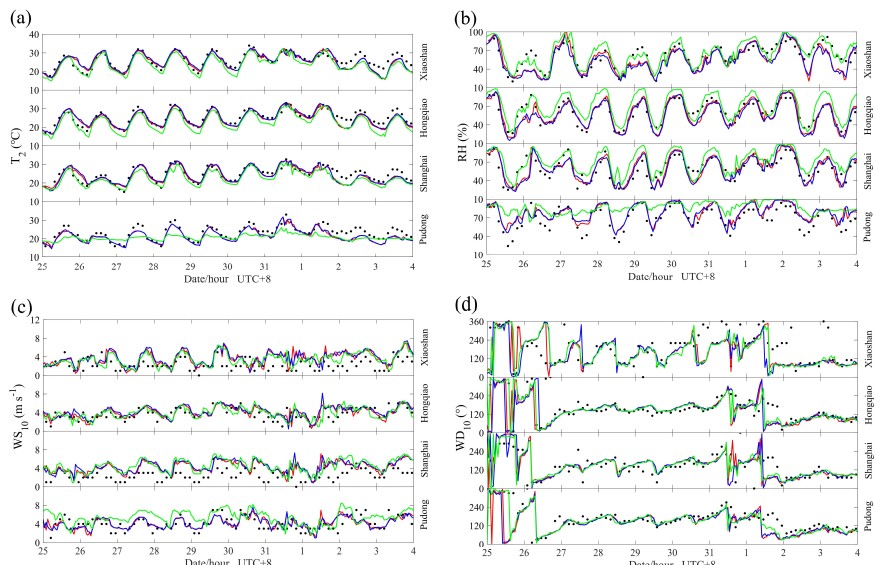


Figure 5. Time series of $T_2$, RH, $WS_{10}$ and $WD_{10}$ for observations and simulations at different

meteorological stations. The black dots are the surface observations. The simulation results of

MODIS_noAH, USGS_noAH and MODIS_withAH are shown in red, green and blue, respectively.

Table 4 lists the statistical metrics in $O_3$, and Figure 6 gives the hourly variations of $O_3$ for

observations and simulations at different sites. With high CORs (the mean CORs are 0.80, 0.81 and

0.80 in MODIS_noAH, USGS_noAH and MODIS_AH, respectively), all the simulations can

reproduce the diurnal variation of $O_3$, which shows that $O_3$ concentration reaches its maximum in

the afternoon and gradually decreases to its minimum in the morning. The magnitude of $O_3$

modeling results is reasonable (Figure 6), but the peak and valley values of $O_3$ simulations are

sometimes differ greatly from the observations, especially the peak value, like Huzhou. This may

be related to the resolution of the emission inventory and the distribution of $O_3$ precursors.

Considering the relatively low mean MB (6.9, -1.6 and 9.0 $\mu g\ m^{-3}$) and mean RMSE (49.3, 46.2 and

49.0 $\mu g\ m^{-3}$), the modeling results of $O_3$ are generally reasonable and acceptable.

**Table 4.** Statistical metrics in $O_3$ between observations and simulations.

| Case | Index | Site |
| --- | --- | --- |


|  |  | CZ | WX | SZ | SH | HZ1 | JX | HZ |
|---|---|---|---|---|---|---|---|---|
|  | $\overline{O}$ | 89.7 | 141.8 | 121.7 | 112.8 | 95.8 | 113.2 | 104.8 |
| MODIS_noAH | $\overline{\overline{S}}$ | 123.2 | 117.6 | 116.2 | 103.4 | 128.1 | 112.5 | 127.5 |
|  | MB | 33.3 | -24.2 | -5.6 | -9.1 | 32.1 | -0.6 | 22.7 |
|  | RMSE | 53.8 | 49.1 | 42.8 | 36.4 | 59.9 | 44.4 | 58.6 |
|  | COR | 0.85 | 0.83 | 0.82 | 0.80 | 0.83 | 0.78 | 0.71 |
| USGS_noAH | $\overline{\overline{S}}$ | 108.1 | 106.8 | 107.1 | 93.8 | 118.6 | 111.0 | 122.5 |
|  | MB | 18.5 | -35.0 | -14.7 | -18.9 | 23.0 | -2.0 | 18.0 |
|  | RMSE | 43.5 | 56.0 | 44.7 | 37.7 | 50.1 | 41.1 | 50.0 |
|  | COR | 0.83 | 0.81 | 0.80 | 0.81 | 0.82 | 0.80 | 0.77 |
| MODIS_AH | $\overline{\overline{S}}$ | 124.5 | 119.8 | 119.1 | 108.0 | 130.3 | 113.7 | 127.8 |
|  | MB | 34.7 | -21.9 | -2.7 | -4.6 | 34.3 | 0.6 | 23.0 |
|  | RMSE | 53.5 | 47.3 | 42.4 | 37.4 | 59.4 | 44.7 | 58.2 |
|  | COR | 0.84 | 0.83 | 0.81 | 0.80 | 0.82 | 0.78 | 0.71 |


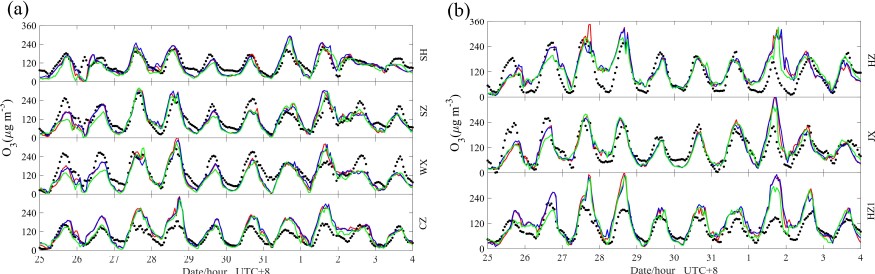


Figure 6. Same as Figure 5, but for $O_3$.

Above all, the WRF-Chem model using our configuration has a good capability in simulating
the meteorology and $O_3$ air quality over the studied region in this study. It is still noteworthy that
the object of inter-comparison between the three numerical experiments is not to determine which
setting is most skillful in reproducing the observations. Rather, it is to diagnose and understand the
differences induced by land-surface forcing and AH, and then to provide valuable insight into the
formation of the $O_3$ pollution episodes.

**3.3 Overall behaviors of $O_3$ and local circulations**
Based on the results of the control simulation (MODIS_noAH), we first give an overall
behavior of $O_3$ and local circulations during the study period. And then the differences induced by
land-surface forcing and AH are discussed via intercomparisons between the numerical experiments.


Thereby, only difference plots between USGS_noAH/MODIS_withAH and MODIS_noAH are
shown in this paper, but the corresponding original plots for USGS_noAH and MODIS_withAH
can be found in supplementary materials (Figure S2-5).
**3.3.1 Spatiotemporal variations of $O_3$**
As show in Figure 7, $O_3$ concentration began to rise around 8:00 local time (LT = UTC + 8 h)
after sunrise, and became noticeable after only 3 hours (Figure 7a and h). During this stage, the
nocturnal residual layer vanished due to the development of the convective boundary layer (Figure
8). The $O_3$-rich air mass in the residual layer was mixed with the $O_3$-poor air mass on the ground,
which enhanced the surface $O_3$ in the morning (Hu et al., 2018). Around 11:00 LT, the convective
boundary layer was established, and high $O_3$ produced by photochemical reactions appeared over
the central YRD and persisted until 18:00 LT (Figure 7b, 7c and 8). The maximum $O_3$ production
was in the middle of the boundary layer (~800 m) instead of at the surface (Figure 8). After sunset,
surface $O_3$ concentrations decreased sharply due to nitrogen oxide (NO) titration. The loss of $O_3$
caused by NO titration almost ceased around 2:00 LT when $O_3$ was at its lowest level of the day
(Figure 7f and g). In general, $O_3$ has a typical diurnal variation with high concentration in the
daytime and low concentration at night. This is consistent with the results in Figure 6, and this rule
of $O_3$ can be applied to most parts of the world. Therefore, the situation during the daytime (We
select 11:00, 14:00, 17:00 and 20:00 LT in this study) should be paid attention to when it comes to
$O_3$ pollution.



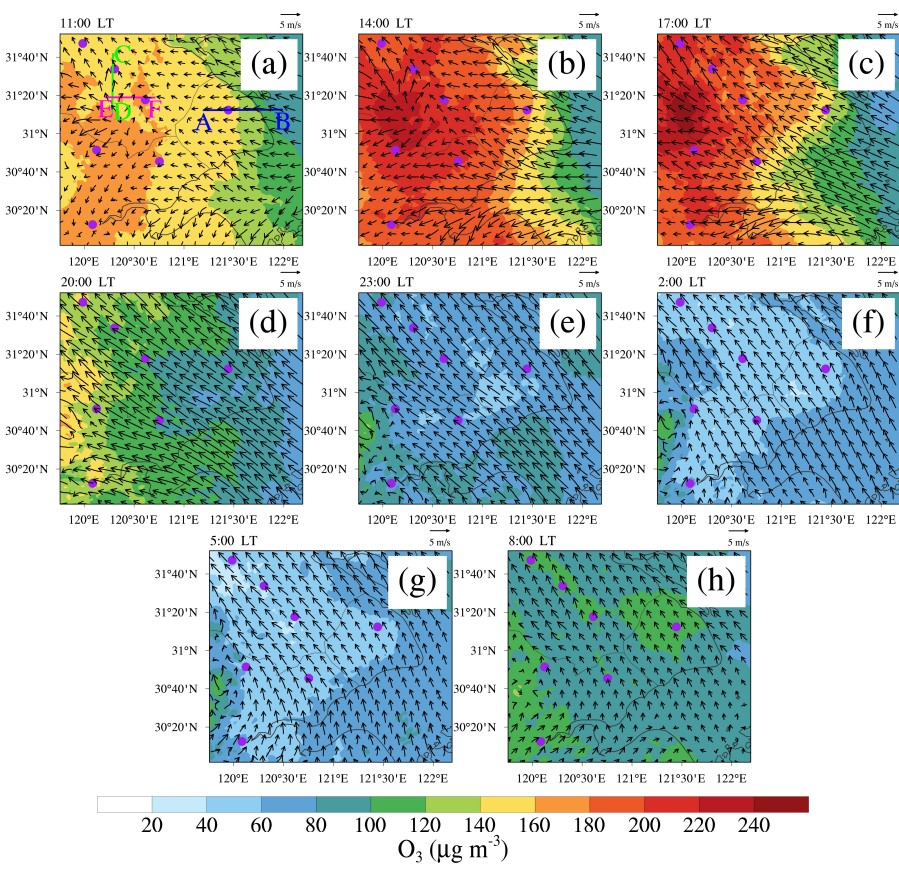

**Figure 7.** Horizontal distributions of $O_3$ and wind at the lowest model level in MODIS_noAH. (a),

(b), (c) and (d) are the results at 11:00, 14:00, 17:00 and 20:00 LT, referring to the daytime. (e), (f),

(g) and (h) are the results at 23:00, 2:00, 5:00 and 8:00 LT, referring to the night. The purple dots

represent the locations of cities (red dots in Figure 1b) in the innermost domain. To obtain general

feature, all results are the average of the study period, and the same for the subsequent results.



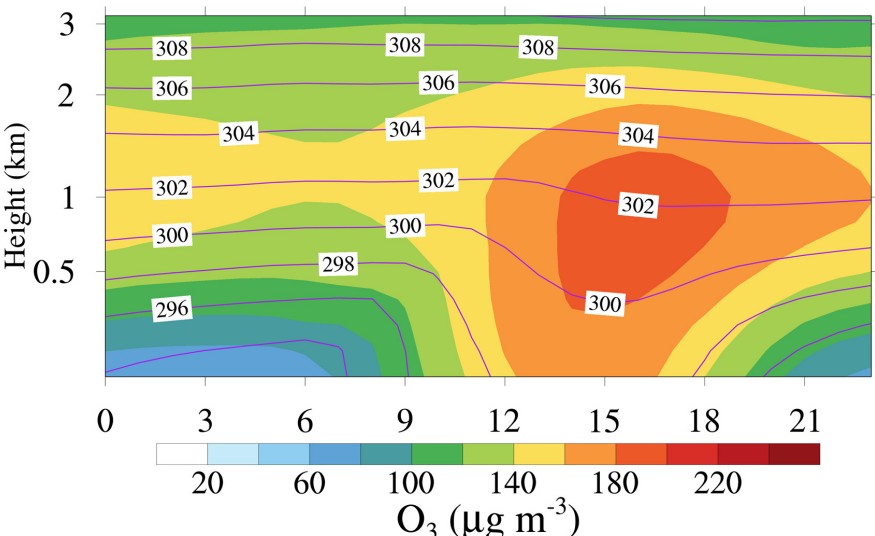

**Figure 8.** Temporal-vertical distribution of $O_3$ and potential temperature over the innermost domain in MODIS_noAH.

### 3.3.2 Sea and lake breezes

As shown in Figure 7a and b, in the areas where the local circulations meet the background dominant winds (the southeast wind), the converging airflows make $O_3$ concentrations higher than those in the surrounding areas. Furthermore, the typical local circulations in the central YRD are the sea and the lake breezes around the Tai Lake. In this study, the sea-breeze usually affected a wide area and lasted a long time, which may be related to the local background field since they are mostly in same direction, and it is difficult to separate the sea-breeze from the southeast wind. The sea-breeze was obvious around 14:00 LT and matured around 17:00 LT, and continuously transported high $O_3$ from coastal to the inland areas during this period (Figure 7b-d). Compared with the sea-breeze, the lake-breeze had a much smaller influencing area and a shorter duration. Around 11:00 LT, the lake-breeze was established. It reached its maximum intensity around 14:00 LT, and then disappeared sharply due to the predominant sea-breeze (Figure 7c). Both the sea and the lake breezes played important roles in the horizontal distributions of $O_3$ in the central YRD.

As the coastline is generally north-south (Figure 1b), the cross sections along line AB depicted in Figure 7a are illustrated to show representative example of the vertical structure of the sea breezes



(Figure 9a-c). The sea-breeze below 500m had already developed by 11:00 LT. A sea-breeze front
was found in front of Shanghai (~121.6°E), with a height of 1.5 km. The speed of sea-breeze
increased around 14:00 LT, which can exceed 5 m s$^{-1}$. The intensified sea-breeze penetrated inland
for a distance of 20-30 km, and the sea-breeze front (~121.4°E) lifted the boundary layer top over
Shanghai up to ~2 km (Figure 9b). Strong sea-breeze swept across the central YRD around 17:00
LT, reducing the $O_3$ concentration near the surface in coastal areas. But the $O_3$ in the mixed layer
still maintained a high level, which can result in an $O_3$-rich reservoir forming in the nocturnal
residual layer (Figure 9c and 8). The penetration of sea-breeze front and its effect on surface $O_3$ can
be also observed in other regions, such as the Pearl River Delta Region (You et al., 2019), Taiwan
(Lin et al., 2007), the Athens basin (Mavrakou et al., 2012) and Paulo (Freitas et al., 2007).

403    As for the lake breezes, the cross sections along line CD (Figure 9d-f) and EF (Figure 9g-i) are

given since the lake is usually inside the land so that the lake breezes can have different directions.
The lake-breeze was established when the surface wind was weak by 11:00 LT (Figure 9d and g)
though it was shallow at that time. Around 14:00 LT, the lake-breeze strengthened. The extension
of the lake-breeze circulation zone can even reach up to 2 km in the vertical dimension. The offshore
flow (~ 2 m s$^{-1}$) of the lake-breeze circulation transported high $O_3$ concentration from urban areas
to the lake, while the onshore flow blew the $O_3$ back to urban areas (Figure 9e and h). Thus, the net
effect is that the lake-breeze "accelerated" the vertical mixing in the boundary layer, resulting in
high concentration of $O_3$ in the lakeside cities. The high surface $O_3$ concentration caused by the lake
breezes has also been confirmed near other lakes, such as the Lake Michigan (Lennartson and
Schwartz, 2002), the Great Lakes (Sills et al., 2011) and the Great Salt Lake (Blaylock et al., 2017).
Finally, the lake-breeze was destroyed by the prevailing southwest wind by 17:00 LT.

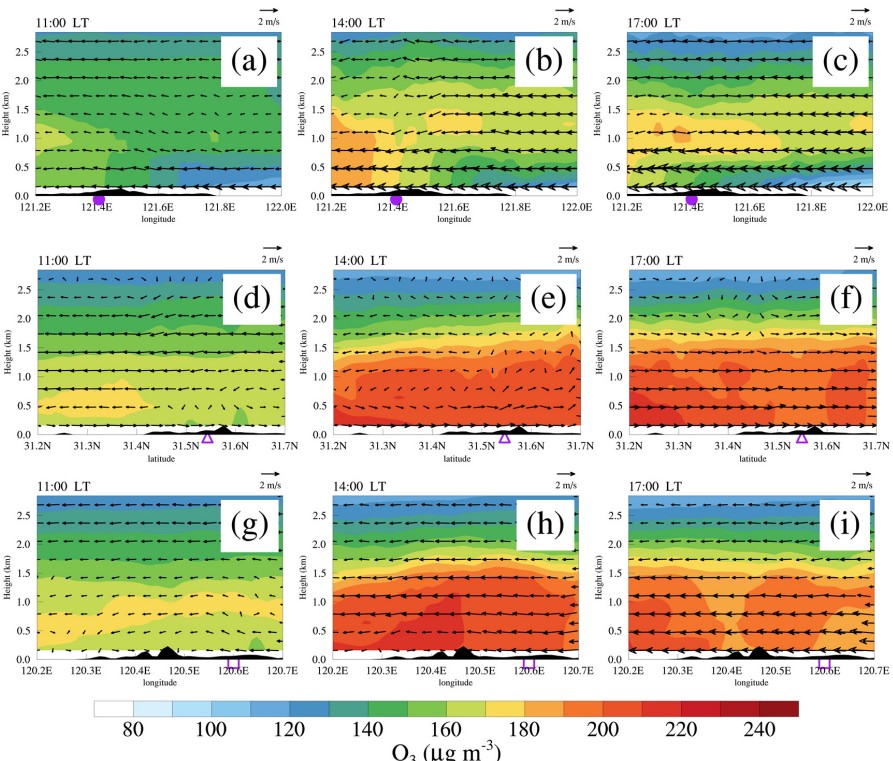

**Figure 9.** Vertical cross sections of $O_3$ and wind for sea-breeze at (a) 11:00, (b) 14:00 and (c) 17:00 LT along the line AB in Figure 7a. (d), (e) and (f) are the same as (a), (b) and (c), respectively, but for lake-breeze along the line CD in Figure 7a. (g), (h) and (i) are also the same as (a), (b) and (c), respectively, but for lake-breeze along the line EF in Figure 7a. The purple dots, triangles and rectangle represent the locations of Shanghai, Wuxi and Suzhou, respectively. The black shaded areas represent the terrain, and the terrain has been multiplied by a factor of 10 when plotting.

**3.4 Impacts of land-surface forcing on meteorology and $O_3$**

**3.4.1 The changes in horizontal direction**

Figure 10 presents the spatial differences of the main factors, including $O_3$, $T_2$, PBLH and $WS_{10}$, between MODIS_noAH and USGS_noAH. Obviously, higher $O_3$ was produced in the MODIS_noAH, indicating that urban expansion will increase surface $O_3$ concentrations. The largest increment of $O_3$ occurred in the afternoon, with a value of 20 μg m$^{-3}$ around 17:00 LT in Changzhou. $T_2$ is directly affected by the land-atmosphere heat fluxes resulting from land-surface forcing. The



spatial pattern of remarkable warming effect for $T_2$ was consistent with the urban-fraction change
(Figure 2a and b), which is that the positive temperature anomaly often appeared in large cities and
their surrounding areas. This positive forcing for $T_2$ is associated with the enhanced surface heating
through upward sensible heat fluxes during the day. In megacities like Shanghai, $T_2$ can increase by
3 ℃. It should be noted that there was a confusing "false" warming at the junction of land and
sea/lake, which was mainly caused by the different treatment of the MODIS-based and USGS land
use classifications at the boundary conditions of land versus water (Figure 2a and b). The change in
PBLH was similar to that in $T_2$, but it was less obvious after sunset around 20:00 LT. This is because
that the warming up of $T_2$ can enhance the vertical air movement in the boundary layer and thereby
increase the PBLH. The maximum positive change of PBLH reached up to 500 m in the urban areas
at noon but downed to 100 m after sunset. The roughness of cities and forest is greater than that of
cropland, so there was a decrease in $WS_{10}$ in the MODIS_noAH (Figure 9m-p), with a maximum
decrease up to 1.5 m s$^{-1}$ in Hangzhou around 17:00 LT.

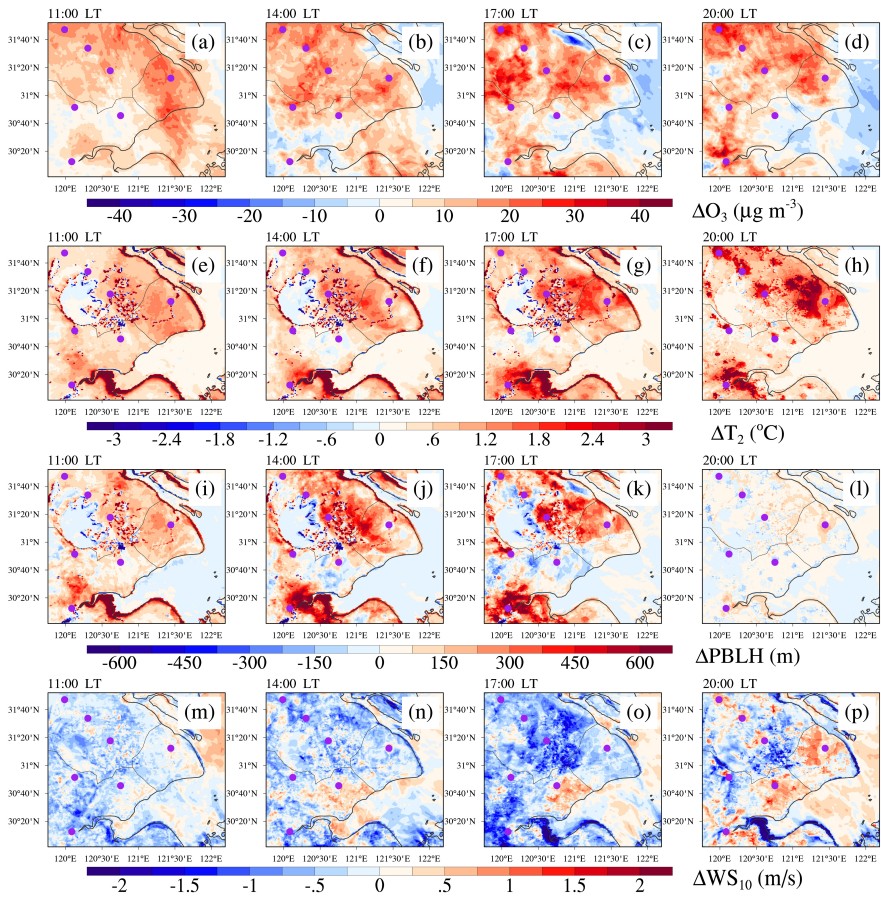

**Figure 10.** Horizontal distributions of the (a-d) $O_3$, (e-h) $T_2$, (i-l) PBLH and (m-p) $WS_{10}$ differences between MODIS_noAH and USGS_noAH (MODIS_noAH – USGS_noAH) at different times (11:00, 14:00, 17:00 and 20:00 LT) of the day. The purple dots represent the locations of cities (red dots in Figure 1b) in the innermost domain.

**3.4.2 The changes in vertical direction**

As shown in Figure 11a-c, the sea-breeze below 500 m increased by 1-2 m s$^{-1}$ due to the existence of the cities which enhanced the temperature contrast between the land and the sea. Strong turbulent mixing and updraft induced by the sea-breeze front promote the development of the urban boundary layer, contributing to elevated $O_3$ levels at surface in the city during the advance of the sea-breeze front inland (Figure 11a and b). When the sea-breeze matured around 17:00 LT, its transport effect reduced the surface $O_3$ concentration of the coastal cities (Figure 9c). However, this



"removal" was weakened because the sea-breeze near the surface was slowed due to the rough urban
surface. Finally, surface $O_3$ of about 10 μg m$^{-3}$ was left compared to the scenario without cities
(Figure 11c).

As for the lake-breeze, it was also enhanced by 1-2 m s$^{-1}$ after the establishment because of the

larger temperature contrast resulting from the cities, just like the sea-breeze (Figure 11e and h). And
the life of the lake-breeze was extended to 17:00 LT (Figure 11f and i) when the city exists. Because
the lake-breeze was conducive to the vertical mixing of the boundary layer and its onshore flow can
blow high concentration of $O_3$ from the lake to the city (Sect. 3.3.2), the urban $O_3$ concentration will
eventually increase, with a maximum of 30 μg m$^{-3}$ in Wuxi at 14:00 LT.

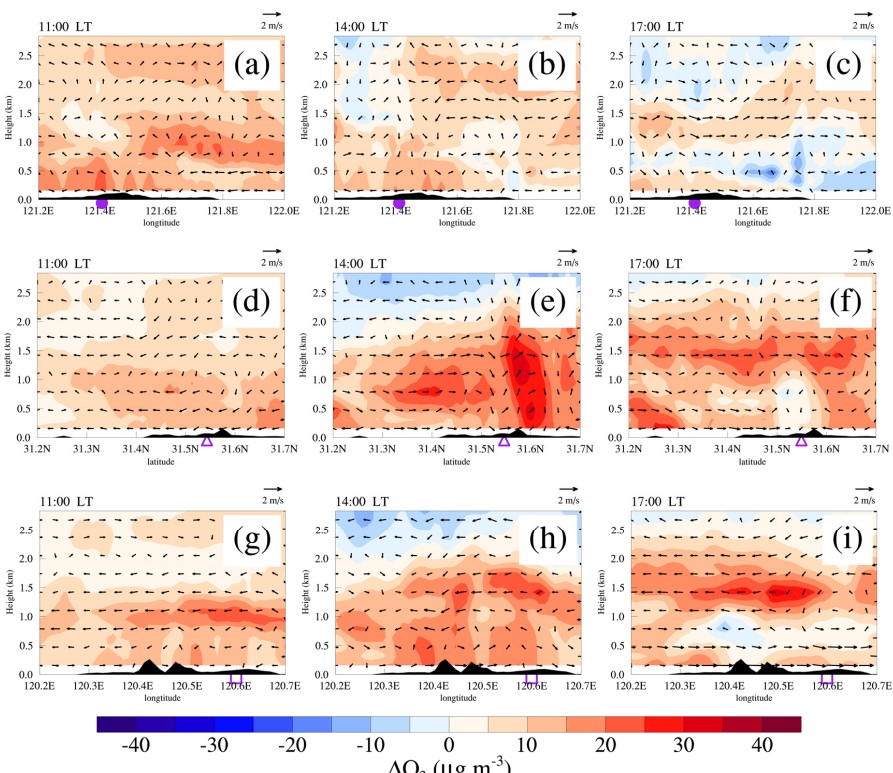


**Figure 11.** Same as Figure 9, but for the differences between MODIS_noAH and USGS_noAH
(MODIS_noAH – USGS_noAH).

**3.4.3 The mechanism of land-surface forcing modulating $O_3$**



Land-surface forcing plays an important role in the evolution of $O_3$ by changing the local
meteorology (meteorological factors and local circulations). Changing land-surface forcing from
USGS to MODIS leads to an increase in $T_2$ by maximum 3 ℃, an increase in PBLH by maximum
500 m and a decrease in $WS_{10}$ by maximum 1.5 m s$^{-1}$ in the YRD, which is comparable to those in
the BTH region (Yu et al., 2012), the PRD region (Li et al., 2014) and the National Capital Region
of India (Sati and Mohan, 2017). And these changes are particularly evident in and around cities.
The elevated air temperature is conducive to the photochemical production of $O_3$, and the well-
developed boundary layer favors the vertical mixing of $O_3$ (Figure 12), which increases the $O_3$
concentration near the surface by maximum 20 μg m$^{-3}$. This change magnitude in $O_3$ is consistent
with the findings reported in Seoul (Ryu et al., 2013b) and Southern California (Li et al., 2019).
Local circulations (the sea and the lake breezes) are also influenced by the land-surface forcing,
chiefly from the urban expansion as the most significant land-surface forcing in the YRD comes
from urban expansion over the past few decades. For the coastal cities, like Shanghai, the larger
temperature contrast induced by cities enhances the sea-breeze below 500 m. As the sea-breeze front
moves inland, it can induce stronger upward air flow that deepens the boundary layer. Thus, high
$O_3$ concentration in the middle of boundary layer can be more easily transported to the surface.
However, the movement of the sea-breeze is slowed due to the rough urban surface after the sea-
breeze matures. The removal of the sea-breeze is then weakened and the surface $O_3$ increases by 10
μg m$^{-3}$. The similar response of the sea breezes to urban expansion as well as its impact on $O_3$ has
been also reported in the PRD region (You et al., 2019) and Paulo (Freitas et al., 2007). For the
lakeside cities, like Wuxi and Suzhou, the lifetime of the lake breezes is extended to the afternoon
due to the existence of the city. The offshore flow of the lake-breeze transports high $O_3$ concentration
in the middle of the boundary layer from the land to the lake, while the onshore flow brings the $O_3$
back to the land, which accelerates the vertical mixing of $O_3$ and can increase the surface $O_3$ by even
30 μg m$^{-3}$. High surface $O_3$ appears when the lake breezes have been established can also be
observed in the Greater Toronto Area (Wentworth et al., 2015) and the Lake Michigan (Abdi-
Oskouei et al., 2020).



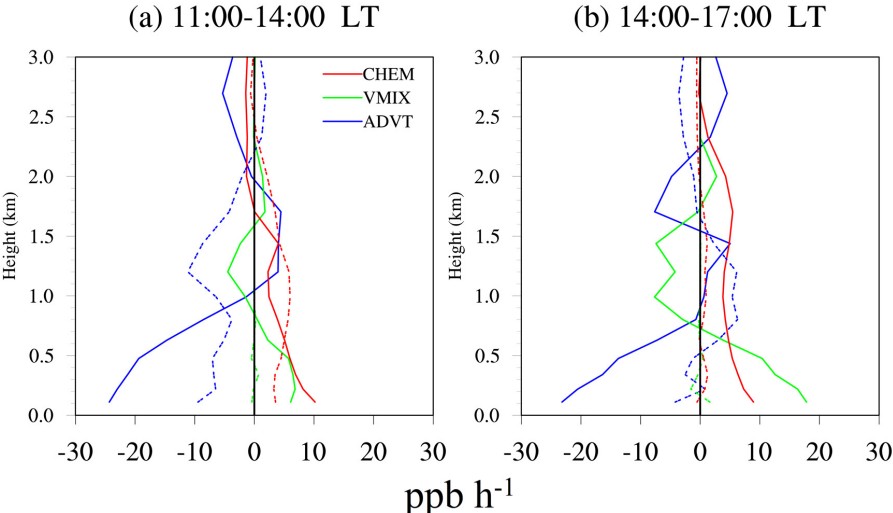


**Figure 12.** Vertical profiles of the changes in individual processes between MODIS_noAH and

USGS_noAH (MODIS_noAH – USGS_noAH) at (a) 11:00-14:00 LT and (b) 14:00-17:00 LT over

Shanghai (solid lines) and Wuxi (dashed lines). CHEM (in red), VMIX (in green) and ADVT (in

blue) represent gas-phase chemical reactions, turbulent mixing and advection transport, respectively.

**3.5 Impacts of anthropogenic heat on meteorology and O$_3$**

**3.5.1 Horizontal changes**

Compared with land-surface forcing, the changes caused by AH are much smaller (Figure 13).

Furthermore, these changes in meteorology and O$_3$ mainly occur in and around cities as there are

more AH emissions in these areas (Figure 3). Surface O$_3$ concentration increased in the urban areas

by about 4 µg m$^{-3}$ in the simulation with adding AH, and this phenomenon was clearer after sunset

(Figure 13d). By adding more surface sensible heat into the atmosphere, the AH fluxes can lead to

an increase in T$_2$ of 0.2 ℃ during the day, with the typical value of 0.42 ℃ in Shanghai. Vertical air

movement in the boundary layer can be enhanced by the warming up of the surface air temperature,

thereby the PBLH will increase as well. According to the simulations, the PBLH increased by about

75 m in the urban areas. With regards to WS$_{10}$, it increased by about 0.3 m s$^{-1}$ in the urban areas,

which is contrary to the decrease in WS$_{10}$ caused by land-surface forcing (Sect. 3.4.1). This is

ascribed to the strengthened urban-breeze circulations induced by the AH fluxes, which is





mentioned in previous studies (Ryu et al., 2013a, b; Xie et al., 2016a, b).

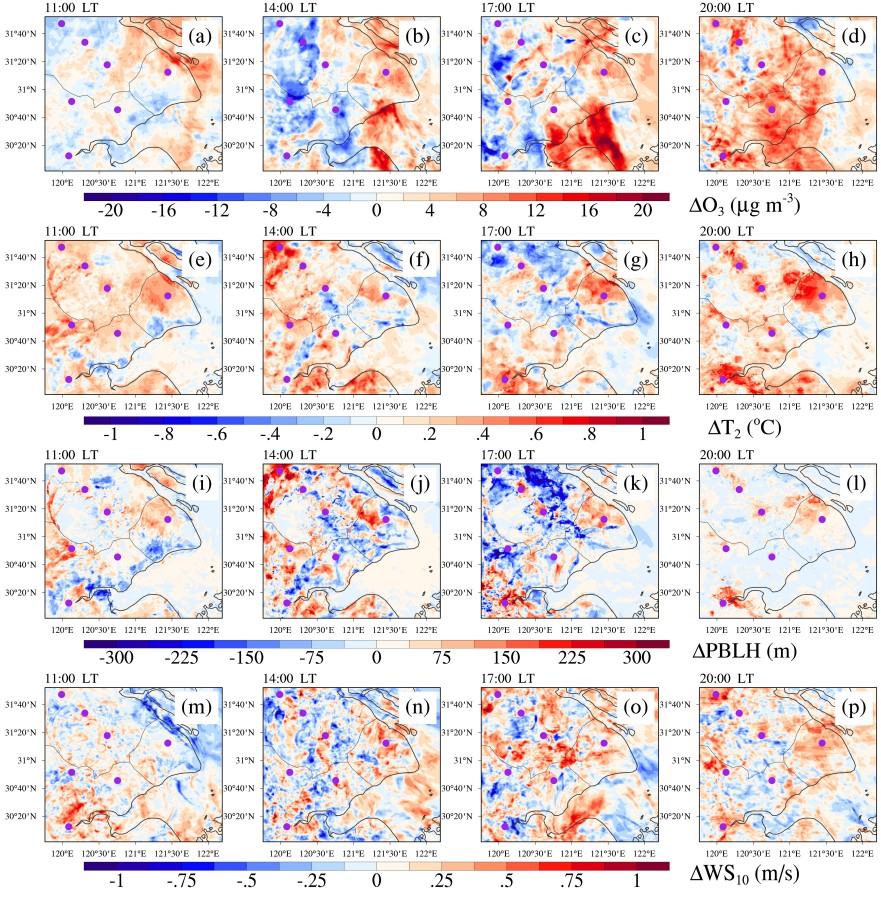


**Figure 13.** Same as Figure 10, but for the differences between MODIS_withAH and MODIS_noAH (MODIS_withAH – MODIS_noAH).


**3.5.2 Vertical changes**

The phenomenon that cities are almost always warmer than their surroundings is known as the
urban heat island (UHI), and the difference between the urban and the rural surface energy balance
can further initiate the UHI circulation. It is clearly seen that an enhanced UHI circulation driven
by AH appeared in the megacity Shanghai around 14:00 LT (Figure 14b). This circulation extended
horizontally 20-30 km from the city center to the urban edge, and vertically to nearly 2 km from the
ground to the top of the urban boundary layer. Under this condition, there was a small increase (4~6





µg m$^{-3}$) in O$_3$ concentrations in the low boundary layer. However, for the lakeside cities, the
enhanced UHI circulation was not visibly noticed, and the O$_3$ concentration in urban areas was
reduced on average, with a maximum of 16 µg m$^{-3}$ in Wuxi around 14:00 LT (Figure 14e). The
lower O$_3$ concentration may be affected by the increased wind on the lake (Figure 13), which was
beneficial to the diffusion and dilution processes. Furthermore, it seems that AH has a limited effect
on local circulations, regardless of the sea or lake breeze, though it play an important role in the
urban-breeze circulations. In our simulation cases, AH does not continuously and significantly affect
any branch of the local circulations like the land-surface forcing.

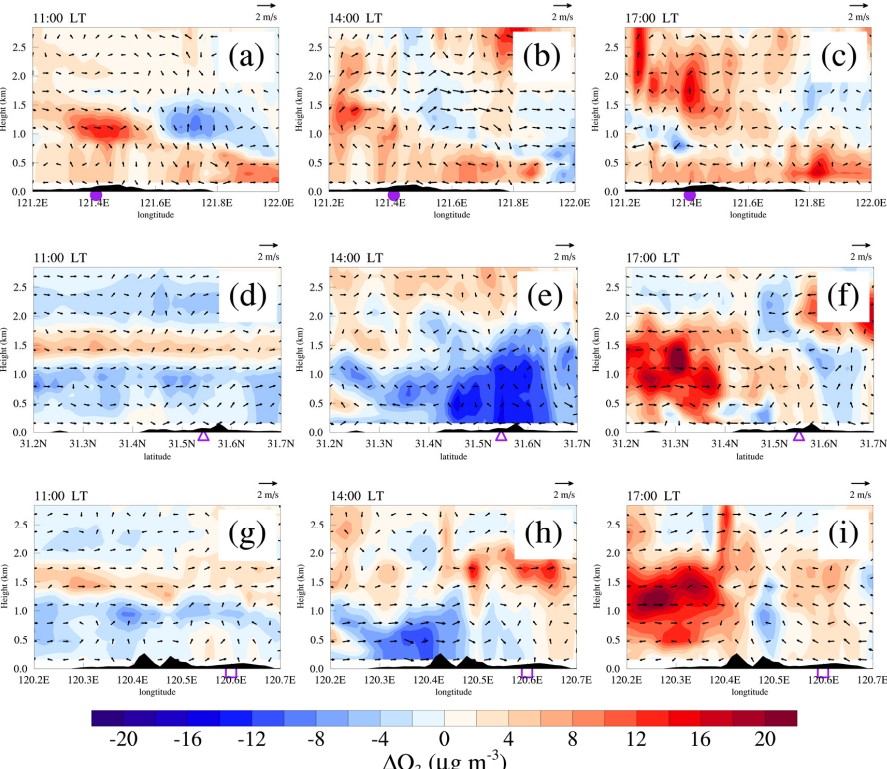


**Figure 14.** Same as Figure 8, but for the differences between MODIS_withAH and MODIS_noAH
(MODIS_withAH – MODIS_noAH).

**3.5.3 The mechanism of anthropogenic heat modulating O$_3$**



AH and land-surface forcing play different roles in meteorology and $O_3$. AH allows the
atmosphere to reserve more energy via the additional sensible heat fluxes, which increases $T_2$ by
about 0.2 °C. Higher temperature is conducive to the development of the convective boundary layer
and can induce stronger upward air movement, which rises the PBLH by about 75 m. In the
convective boundary layer, the atmosphere is associated with turbulent motions, and is unstable.
Together with the urban-breeze circulations enhanced by AH, $WS_{10}$ can increase by 0.3 m s$^{-1}$. These
findings are comparable to the values estimated in other cities all around the world, such as
Philadelphia in the United States (Fan and Sailor, 2005), Winnipeg in Canada (Ferguson and
Woodbury, 2007), Berlin in German (Menberg et al., 2013) and Tokyo in Japan (Dhakal and Hanaki,
2002). It is noteworthy that the abovementioned changes mainly appear in large cities and their
surrounding areas, where AH emission centers are located. And these changes eventually caused an
increase in surface $O_3$ concentration by about 4 μg m$^{-3}$. Additionally, though AH can play an
important role in urban-breeze circulations, it may not be powerful enough to affect the local
circulations such as the sea and the lake breezes.

**4 Summary and conclusions**
Land-surface forcing related to the urban expansion and AH release from human activities can
change the meteorology (meteorological factors and local circulations) and thereby affect $O_3$ air
quality in and around cities. In this study, the YRD region, a highly urbanized place with sever $O_3$
pollution and complex geography, is selected to discuss this issue. Firstly, we briefly describe the
general characteristics of $O_3$ pollution in the YRD based on the surface observations. Secondly, we
simulate a representative case using WRF-chem and evaluate the model performance by comparing
with the observational data. Finally, the response of meteorology as well as $O_3$ to land-surface
forcing and AH are investigated from the model results. The main findings are listed as below:
(1) Regional $O_3$ pollution occurs frequently in the YRD (~ 26 times per year). Like other
regions, these $O_3$ pollution episodes mainly occur in warm season (April to October) under calm
conditions characterized by high temperature (over 20 °C), low relative humidity (less than 80%),
light wind (less than 3 m s$^{-1}$) and shallow cloud cover (less than 5). In this case, the local circulations
induced by thermal differentiation tend to develop and will have an important impact on the
distribution of $O_3$.


(2) By updating the land-use data from USGS to MODIS, we find an increase in $T_2$ by
maximum 3 °C, an increase in PBLH by maximum 500 m and a decrease in $WS_{10}$ by maximum 1.5
m s$^{-1}$ in the YRD, which is comparable to those in the BTH region (Yu et al., 2012), the PRD region
(Li et al., 2014) and the National Capital Region of India (Sati and Mohan, 2017). The higher
temperature and PBLH elevate the $O_3$ level by maximum 20 μg m$^{-3}$ via the photochemical and the
vertical mixing processes, respectively. For changes in local circulations, the sea-breeze below 500
m is enhanced due to larger temperature contrast induced by the urban expansion. During the
advance of the sea-breeze front inland, the upward air flow in front of the front is conducive to the
vertical mixing of $O_3$. When the sea-breeze is well formed in the late afternoon, further progression
inland is stalled on account of the rough urban surface. The transport of high $O_3$ from coastal to the
inland areas is weakened and thereby $O_3$ can be 10 μg m$^{-3}$ higher in the case with cities than without.
The similar results have been also reported in the Paulo (Freitas et al., 2007) and the PRD region
(You et al., 2019). With respect to the lake breezes, its lifetime will be extended from the noon to
the afternoon because of the urban expansion. Since the net effect of the lake-breeze is to accelerate
the vertical mixing in the boundary layer, the surface $O_3$ can increase as much as 30 μg m$^{-3}$
influenced by the lake-breeze. Similar phenomenon also be observed in the Greater Toronto Area
(Wentworth et al., 2015) and the Lake Michigan (Abdi-Oskouei et al., 2020).
(3) The changes caused by AH are different from land-surface forcing. These changes are
relatively small and mainly appear around the cities where there are large AH emissions. Through
regulating the land-atmosphere heat fluxes, $O_3$, $T_2$, PBLH and $WS_{10}$ increases by about 4 μg m$^{-3}$,
0.2 °C, 75 m and 0.3 m s$^{-1}$ under the effect of the additional sensible heat fluxes induced by AH.
The magnitudes of these changes are consistent with the values estimated in other cities all around
the world, including Philadelphia in the United States (Fan and Sailor, 2005), Winnipeg in Canada
(Ferguson and Woodbury, 2007), Tokyo in Japan (Dhakal and Hanaki, 2002) and Berlin in German
(Menberg et al., 2013). Additionally, our results show that AH may have a quite limited impact on
local circulations, such as the sea and the lake breezes. But the urban-breeze circulations in and
around big cities are sensitive to AH inputs, which can further affect the urban air pollutants.
Estimating the impacts of land-surface forcing and AH on urban climate and air quality is a
complex but necessary issue as these two are important manifestations of urbanization. Although
our study only focuses on the YRD region, most of the results can be supported by previous studies



that conducted in other region around the world. Thus, our work may provide valuable insight into
the formation of $O_3$ pollution in those rapidly developing regions with unique geographical features.

***Data Availability Statement.***
Air quality monitoring data were acquired from a mirror of data from the official NEMC real-time
publishing platform (https://quotsoft.net/air/). Meteorological data were issued by the NCDC
(ftp://ftp.ncdc.noaa.gov/pub/data/noaa/isd-lite/). The FNL meteorological data were acquired from
NCEP (https://doi.org/10.5065/D6M043C6/). These data can be downloaded for free as long as you
agree to the official instructions.

***Author contributions.***
CZ and MX had the original ideas, designed the research, collected the data and prepared the original
draft. CZ did the numerical simulations and carried out the data analysis. MX acquired financial
support for the project leading to this publication.

***Acknowledgements.***
This work was supported by the National Key Research and Development Program of China
(2018YFC0213502, 2018YFC1506404). We are grateful to MEPC for the air quality monitoring
data, to NCDC for the meteorological data, to NCEP for global final analysis fields and to Tsinghua
University for the MEIC inventories. The numerical calculations have been done on the Blade
cluster system in the High Performance Computing and Massive Data Center (HPC&MDC) of
School of Atmospheric Sciences, Nanjing University. We also thank the constructive comments and
suggestions from the anonymous reviewers.

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
