# Peer review of "Land use and anthropogenic heat modulate ozone by meteorology"

_Atmospheric Chemistry and Physics, 2021_

## Referee Comment (RC2)

**Comments on "Land-surface forcing and anthropogenic heat modulate ozone by meteorology: A perspective from the Yangtze River Delta region"**

This paper describes the general characteristics of regional O3 pollution in the Yangtze River Delta (YRD) region, a highly urbanized place with complex geography. The impacts of land-surface forcing and anthropogenic heat (AH) on meteorological factors, local circulations and O3 are investigated by using the WRF-Chem model. This is an interesting topic to diagnose the changes in local circulations affected by land-surface forcing/AH and their effects on O3 because these elements are usually at different scales. From this paper, the interactions of these multi-scale local circulations seem to play an important role in O3 pollution, and this may be an important supplement to current research on related topics. Thus, the manuscript can be considered to be published in ACP after making revisions as follows:

**General comments**

1. I suggest replacing the "land-surface forcing" in the title as well as the corresponding place in the text with "land use".

2. Section 3.2.1, since the subsequent results are based on this case, I believe that a more detailed description is needed to make sure the case is in a calm weather. This is an important prerequisite for the formation of local circulations. In fact, plenty of materials, like the time series of meteorological factors, have been given in the section of model evaluation, but should be condensed here.

**Specific comments**

1. Lines 359-361. The titration of NO does not terminate, and surface O3 seems to be lowest in the early morning (Figure 6).

2. Lines 395-397. "... the sea-breeze front lifted the boundary layer ...", the development of the boundary layer mainly depends on solar radiation, although some factors do affect the boundary layer height.

3. Figure 7 and 9. O3 concentration on the lake is higher than that in the city during the day. Why? Will this affect the lakeside cities?

4. Section 3.4.2. Only the sea breeze was discussed in this section. However, both the offshore and onshore flows should be considered when we discuss circulations.

5. Wind arrows in Figure 11 and 14 are too small to identify. Please improve the figure presentations with better quality.

**Technical corrections**

- 1. "MODIS\_withAH" and "MODIS\_AH" should be the same, please choose any one of them.
- 2. Line 258, "is" -> "are".
- 3. Line 277, "provide" -> "provides".
- 4. A few typos, grammatical and syntactic mistakes need to be corrected.

---

## Author Comment (AC1)

**A point to point response to the reviewers' comments**

Thank you for the reviewers' comments on our manuscript entitled "Land-surface forcing and anthropogenic heat modulate ozone by meteorology: A perspective from the Yangtze River Delta region" (acp-2021-619). Those constructive comments are all valuable for revising and improving our manuscript, as well as the important guiding significance to our researches. We have studies those comments carefully and have made correction which we hope to meet with approval. Here are point to point responses (in blue colored). Accordingly, we also revised the manuscript (in red colored). The line numbers in response are obtained from the unmarked manuscript, in which all revisions have been accepted.

**Anonymous Referee #1:**

**General comments**

This manuscript aims to examine the impact of different land cover (LC) data sets and anthropogenic heat on meteorology and air quality as well as on one of the most populated areas in the world. The manuscript is clear and well written. Despite that, I think this manuscript needs considerable improvement to be considered a novel and meaningful contribution to the existing literature on the subject.

Response: We would like to express our great appreciation to you for the valuable and constructive comments on our manuscript. We have tried our best to revise our manuscript based on the following comments.

**Specific comments**

1. Although the authors discuss in detail the impact of land cover, there are a lot of previous studies which study the impact of land cover datasets on meteorology and air quality. Therefore, this manuscript is just confirming what has been seen in previous papers (I would also like to mention Jiang et al., 2008, Xuemei et al 2009, Park et al., 2014, Fu and Liao, 2014). USGS is an outdated dataset, and the resulting model outputs are somehow expected, as the authors stated in line 427 "Obviously, higher $O_3$ was produced in the MODIS_noAH, indicating that urban expansion will increase surface $O_3$ concentrations." Since the authors did not perform a long-term analysis to see

the impact of urbanization (MODIS LU) vs pre-urbanization (USGS LU) on meteorology and air quality, but rather a short-term simulation in which they evaluate the model against measurements, I do not see why they are using an obsolete dataset for the present-day simulation. It would be nice to analyze the impact of MODIS vs From-GLC_2015.

Response: Thanks for the constructive comment. We agree that previous studies have extensively discussed the changes in meteorology and surface ozone caused by urban expansion and anthropogenic heat. However, as we mentioned in the introduction, these studies rarely focused on the interactions between atmospheric circulations at different scales. The coincidence with previous work gives us strong confidence to investigate the responses of local/urban circulations to urban expansion/anthropogenic heat and their effects on surface ozone. We find that both ozone pollution episodes and local circulations tend to occur under clam weather conditions. Even if we "randomly" select an ozone pollution episode, the phenomenon that local circulations have an important influence on the distribution of surface ozone still holds true. What's more, the emission intensity changes little in such a short period of time, and thereby the influences of the emission rate are not involved in this study. According to the results of our paper, changes in land use induced by urban expansion alter the meteorological factors and local circulations (the sea and the lake breezes) throughout the YRD, thereby affecting the surface ozone. On the other hand, anthropogenic heat can only affect environment around cities. These results provide valuable insight in inter-regional/regional prevention and control for ozone since ozone pollution is usually regional.

There are two reasons that we choose MODIS vs USGS instead of MODIS vs From-GLC_2015. One is that MODIS and USGS are the two default land use classifications of WRF. Using these two can ensure the correctness of our results. The other is that there is a clear difference for urban areas between MODIS and USGS, while MODIS and From-GLC_2015 do not (Figure 2 in the manuscript). The urban fraction in MODIS is close to that in From-GLC_2015, which can generally refer to today's distribution of cities. Thus, using MODIS and USGS can well achieve the purpose of studying the impact of urban expansion on local circulations. In the revised manuscript, we have emphasized that USGS represents the distribution of cities in the late 1980s (lines 151-154), and MODIS can generally refer to current cities distribution (lines 156-160).

2. On a related note, the changes in predicted ozone are solely driven by changes in meteorology due to the use of different LC datasets. Have the authors considered updating the input MEGAN2 file for their simulation to match the LC data? The distribution of PFTs from the standard MEGAN2 input data are those from 2001. Therefore, the changes in the BVOCs emissions are exclusively influenced by changes in the meteorology since the PFTs remain unchanged throughout the simulations.

Response: Thanks for the constructive comment. In this paper, we are currently paying more attention to the meteorological changes caused by urbanization to urban ozone pollution episodes. Thus, we use the same surface biogenic emission rates for different land use scenarios. Following the previous work (Li et al., 2014, 2017), we have updated the necessary input file in MEGAN. Future research can be carried out in quantifying the contribution of BVOCs affected by meteorology to ozone, but it is beyond the scope of this paper. We have added this point to the manuscript. Please see lines 198-202 in the revised manuscript. Thanks again for your reminder and constructive comment.

References

Li, M., Song, Y., Huang, X., Li, J., Mao, Y., Zhu, T., Cai, X., and Liu, B.: Improving mesoscale modeling using satellite-derived land surface parameters in the Pearl River Delta region, China, Journal of Geophysical Research: Atmospheres, 119, 6325-6346, 10.1002/2014jd021871, 2014.

Li, M., Wang, T., Xie, M., Zhuang, B., Li, S., Han, Y., Song, Y., and Cheng, N.: Improved meteorology and ozone air quality simulations using MODIS land surface parameters in the Yangtze River Delta urban cluster, China, Journal of Geophysical Research: Atmospheres, 122, 3116-3140, 10.1002/2016jd026182, 2017.

**Technical corrections**

1. Line 101: "the" instead of "The".

Response: Thanks for the constructive comment. The word "The" in the original manuscript is revised to "the". Please see line 98 in our new manuscript.

2. Line 152: delete etc.

Response: Thanks for the constructive comment. In the revised manuscript, we have deleted the word ", etc". Please see lines 142-143.

3. Line 184: briefly describe how the AH fluxes have been added.

Response: Thanks for the constructive comment. We rewrite "Sect.2.3 Anthropogenic heat flux modeling", in which we have briefly described how to add AH fluxes to the model. Please see lines 171-175 in the revised manuscript.

Within the Single Layer Urban Canopy Model (SLUCM), the default parameterization for AH is described by the following algorithm:

$$SH = F_V \cdot SH_V + F_U \cdot (SH_U + AH_{fixed}), \qquad (1)$$

where SH is the total sensible heat flux in a grid. $F_V$ and $SH_V$ are the fractional coverage and the sensible heat flux of vegetation, respectively. $F_U$ and $SH_U$ are those of urban surfaces. $AH_{fixed}$ represents the fixed AH value for all urban areas. We modify Eq. (1) by incorporating the inhomogeneous AH data ($Q_F$) as follows:

$$SH = F_V \cdot SH_V + F_U \cdot (SH_U + Q_F), \qquad (2)$$

In this study, the $Q_F$ were calculated based on the statistics data of energy consumption of China in 2016.

4. Line 217: why are you using 88 hours for spin-up?

Response: Thanks for the constructive comment. Usually, a spin-up time of several hours will be set to reduce the effect of errors in initial conditions when we simulate meteorological fields. However, it takes a longer spin-up time when doing air quality simulations. In our simulations, we find that the simulation results are acceptable from the fourth day (Figure R1). Coupled with the time zone conversion, the final spin-up time is 88 (24*4-8=88) hours.

[Figure]

**Figure R1.** Horizontal distributions of O$_3$ and wind at the lowest model level in MODIS_noAH at 8:00 LT on (a) May 22, (b) May 23, (c) May 24 and (d) May 25, 2017. Our model started at 00:00 UTC (8:00 LT) on May 21, 2017.

5. Line 238: define adverse weather conditions.

Response: Thanks for the constructive comment. To avoid using the puzzling phrase, we replace "Under adverse weather conditions" with "On cloudless sunny days". Please see line 233 in the revised manuscript.

6. Line 251: define Meiyu since the reader might not know the local name of a frontal system.

Response: Thanks for the constructive comment. After careful consideration, it may not be necessary to explain the reason for the monthly variations in temperature and relative humidity.

Thus, we remove the sentence "This may be related to the Meiyu in June, and the hot weather in July as the YRD is usually dominated by the western Pacific subtropical high after Meiyu." in the revised manuscript.

7. Lines 255-261: It seems that the authors made these statements using only a visual analysis of the data depicted in Fig.4. It would be nice to see the correlation between meteorological variables and MDA8 $O_3$.

Response: Thanks for the constructive comment. We have calculated the correlation coefficients between temperature, relative humidity, cloud cover, wind speed and MDA8 $O_3$, with the values of 0.12, -0.34, -0.15 and 0.04, respectively. We then add this to the revised manuscript, please see lines 247-251.

Due to the different locations of weather stations and monitoring stations (Figure 1b), the correlation between meteorological variables and MDA8 $O_3$ is usually not high. Especially for the wind speed, the correlation seems to be untenable. This may be related to the small change in weak wind due to the existence of start-up wind speed (Figure 5c). Nevertheless, the positive (negative) correlation between MDA8 $O_3$ and temperature (relative humidity and cloud cover) still indicates that $O_3$ pollution episodes tend to occur on days characterized by high temperature, low relative humidity and cloudless sky. Specifically, as shown in Figure 4, more than half of $O_3$ pollution episodes involve temperature higher than 20 ℃, relative humidity lower than 80%, cloud cover less than 5 okta and wins speed less than 3 m s$^{-1}$. And this also applies to different months in the YRD.

8. Line 300: Have the authors consider using the "topo_wind" option to turn on the surface wind correction?

Response: Thanks for the constructive comment. We rechecked the "namelist.input" file, and find that we did not turn on the "topo_wind" option. We take your suggestion, and turn on this option on the basis of the control simulation (MODIS_noAH). To evaluate the model performances, Table R1 lists the statistical metrics in meteorological variables and ozone. Though this option does affect both meteorological factors and ozone, the impacts seem to be relatively limited. With regard to the

wind fields, this option cannot solve the problem of overestimation of wind speed in WRF-Chem under calm weather conditions. Therefore, we do not further test this option in USGS_noAH and MODIS_AH due to computer cost.

**Table R1.** Comparisons between the simulations and observations.

| Case | Index | Variables | | | | |
|---|---|---|---|---|---|---|
| | | $T_2$ (°C) | $RH_2$ (%) | $WS_{10}$ (m/s) | $WD_{10}$ (°) | $O_3$ (µg/m³) |
| | $\bar{O}$ | 24.8 | 62.1 | 3.2 | 162.6 | 111.4 |
| topo_wind =0 (off) | $\bar{S}$ | 23.7 | 62.9 | 4.0 | 145.7 | 118.3 |
| | MB | -1.0 | 0.7 | 0.8 | -18.3 | 6.9 |
| | RMSE | 2.2 | 12.1 | 1.7 | 81.2 | 49.3 |
| | COR | 0.87 | 0.80 | 0.41 | 0.49 | 0.80 |
| topo_wind =1 (Jimenez method) | $\bar{S}$ | 23.7 | 63.0 | 4.0 | 147.2 | 118.5 |
| | MB | -1.1 | 0.9 | 0.8 | -16.9 | 7.1 |
| | RMSE | 2.3 | 12.3 | 1.7 | 80.9 | 49.2 |
| | COR | 0.87 | 0.83 | 0.42 | 0.50 | 0.80 |
| topo_wind =2 (UW method) | $\bar{S}$ | 23.7 | 63.2 | 4.0 | 142.3 | 119.0 |
| | MB | -1.0 | 1.1 | 0.8 | -21.8 | 7.5 |
| | RMSE | 2.2 | 11.9 | 1.7 | 85.1 | 48.1 |
| | COR | 0.88 | 0.84 | 0.45 | 0.44 | 0.81 |

9. Figure 7: please use filled dots showing the measurements instead of purple dots if the measurements are available. It is difficult to spot the lines AB, CD, and EF.

Response: Thanks for the constructive comment. In the revised Figure 7, we replace the purple dots with filled dots showing the measurements to show the city locations. Please see line 364. Also, we have done the same for the Figure S2 and S5. Please see line 16 and 24 in the supporting information.

As for the problem that it is difficult to spot the lines AB, CD and EF in Figure 7, we decide to draw these three lines in Figure 2 instead of Figure 7. Figure 2 shows the land cover maps in the innermost domain using USGS and MODIS. In this case, these three lines can be easier to identify due to their color and thickness. Please see line 162 for the new Figure 2 in the revised manuscript.

10. Figure 8: Add the modeled PBLH on the plot to sustain the sentence "The maximum $O_3$ production was in the middle of the boundary layer (~800 m) instead of at the surface".

Response: Thanks for the constructive comment. We have added the modeled planetary boundary layer height (PBLH) on the revised Figure 8. Please see Figure 8 in line 371. Moreover, we only average the models results of the cities instead of the innermost domain since we are more concerned about cities and their surrounding areas. Although researchers have reported that convection boundary layer is conducive to the photochemical production of ozone (Tang et al., 2017, 2021). The conclusion that "The maximum $O_3$ production was in the middle of the boundary layer (~800 m) instead of at the surface" cannot be directly derived from Figure 8 due to fully mixture of ozone in the entire boundary layer. Therefore, we delete this sentence in our new manuscript.

References

Tang, G., Zhu, X., Xin, J., Hu, B., Song, T., Sun, Y., Zhang, J., Wang, L., Cheng, M., Chao, N., Kong, L., Li, X., and Wang, Y.: Modelling study of boundary-layer ozone over northern China - Part I: Ozone budget in summer, Atmospheric Research, 187, 128-137, 10.1016/j.atmosres.2016.10.017, 2017.

Tang, G., Liu, Y., Huang, X., Wang, Y., Hu, B., Zhang, Y., Song, T., Li, X., Wu, S., Li, Q., Kang, Y., Zhu, Z., Wang, M., Wang, Y., Li, T., Li, X., and Wang, Y.: Aggravated ozone pollution in the strong free convection boundary layer, Sci Total Environ, 788, 147740, 10.1016/j.scitotenv.2021.147740, 2021.

11. Line 363: "we" instead of "We".

Response: Thanks for the constructive comment. The word "We" in the original manuscript is revised to "we". Please see line 360 in our new manuscript.

12. Lines 473-499: Some sentences belong to the introduction, or they are repeating the same information presented in the introduction.

Response: Thanks for the constructive comment. We have deeply reorganized the introduction, Sect.3.4.3 (the mechanism of land use modulating $O_3$), Sect. 3.5.3 (the mechanism of anthropogenic heat modulating $O_3$) and summary to reduce the repetitive content of these chapters, which makes our manuscript much more condensed.

Jiang et al., 2008, doi:10.1029/2008JD009820

Xuemei et al. 2009, DOI:10.1007/s00376-009-8001-2

Park et al, 2014, https://doi.org/10.5194/acp-14-7929-2014

Yu and Liao, 2014, https://doi.org/10.3402/tellusb.v66.24987

Thanks for the recommended literatures. We have learned a lot from these literatures, all the recommended literatures are citied in our new manuscript.

---

## Author Comment (AC2)

**A point to point response to the reviewers' comments**

Thank you for the reviewers' comments on our manuscript entitled "Land-surface forcing and anthropogenic heat modulate ozone by meteorology: A perspective from the Yangtze River Delta region" (acp-2021-619). Those constructive comments are all valuable for revising and improving our manuscript, as well as the important guiding significance to our researches. We have studies those comments carefully and have made correction which we hope to meet with approval. Here are point to point responses (in blue colored). Accordingly, we also revised the manuscript (in red colored). The line numbers in response are obtained from the unmarked manuscript, in which all revisions have been accepted.

**Anonymous Referee #2:**

This paper describes the general characteristics of regional $O_3$ pollution in the Yangtze River Delta (YRD) region, a highly urbanized place with complex geography. The impacts of land-surface forcing and anthropogenic heat (AH) on meteorological factors, local circulations and $O_3$ are investigated by using the WRF-Chem model. This is an interesting topic to diagnose the changes in local circulations affected by land-surface forcing/AH and their effects on $O_3$ because these elements are usually at different scales. From this paper, the interactions of these multi-scale local circulations seem to play an important role in $O_3$ pollution, and this may be an important supplement to current research on related topics. Thus, the manuscript can be considered to be published in ACP after making revisions as follows:

Response: We would like to thank the referee for the valuable and affirmative comments of our manuscript. We carefully revise our manuscript based on the following comments.

**General comments**

1. I suggest replacing the "land-surface forcing" in the title as well as the corresponding place in the text with "land use".

Response: Thanks for the constructive comment. We agree that "land use" is more accurate and common than "land-surface forcing", which is more in line with our topic. We have replaced the phrase "land-surface forcing" with "land use" in the full text.

2. Section 3.2.1, since the subsequent results are based on this case, I believe that a more detailed description is needed to make sure the case is in a calm weather. This is an important prerequisite for the formation of local circulations. In fact, plenty of materials, like the time series of meteorological factors, have been given in the section of model evaluation, but should be condensed here.

Response: Thanks for the constructive comment. We take your suggestion and deeply reorganize this section. In the revised manuscript, we add a table containing detailed information about ozone concentrations and meteorological variables (2-m air temperature, relative humidity, 10-m wind speed and cloud cover) during this ozone pollution episode. These contents are also described in the corresponding text. Please see lines 267-281 in the revised manuscript.

During this ozone pollution episode, the means of MDA8 $O_3$, 2-m air temperature, relative humidity, 10-m wind speed and cloud cover in the Yangtze River Delta region were 182.1 μg m$^{-3}$, 26.4 °C, 58.6%, 2.8 m s$^{-1}$ and 4.2 okta, respectively. The values of these meteorological variables meet the general standard that the temperature exceeds 20 °C, the relative humidity is less than 80%, the wind speed is less than 3 m s$^{-1}$ and the cloud cover is less than 5 okta (Sect. 3.1). Furthermore, the weather pattern was dominated by high pressure/uniform pressure field (Table S1) during this smog episode. Therefore, this case is in a calm weather, which is conductive to the formation of ozone pollution and local circulations.

**Specific comments**

1. Lines 359-361. The titration of NO does not terminate, and surface $O_3$ seems to be lowest in the early morning (Figure 6).

Response: Thanks for the constructive comment. We are deeply sorry for the confusing sentence "The loss of $O_3$ caused by NO titration almost ceased around 2:00 LT when $O_3$ was at its lowest level of the day" here. This sentence is revised as "surface $O_3$ concentration generally decreased due to nitrogen oxide titration, and reached its minimum in the early morning" in the new manuscript (lines 356-358), hoping this statement is accurate.

2. Lines 395-397. "… the sea-breeze front lifted the boundary layer …", the development of the boundary layer mainly depends on solar radiation, although some factors do affect the boundary layer height.

Response: Thanks for the constructive comment. The sentence "The intensified sea-breeze penetrated inland for a distance of 20-30 km, and the sea-breeze front lifted the boundary layer top over Shanghai up to ~ 2 km." is a clerical error. The original intention should be that "The sea breeze front moved inland for a distance of 20-30 km, and was elevated to ~ 2 km". We have corrected this in the new manuscript, please see lines 391-392. Thanks again for your reminder.

3. Figure 7 and 9. $O_3$ concentration on the lake is higher than that in the city during the day. Why? Will this affect the lakeside cities?

Response: Thanks for the constructive comment. Figure R2 shows the distribution of different processes to $O_3$ concentration in the control simulation (MODIS_noAH). Although stronger photochemical reactions are found on the land (Figure R2a and b), the amount of $O_3$ deposition on the lake is much smaller than that on the land (Figure R2d and e). The small dry deposition velocity on the water is also mentioned in Park et al. (2014), an important literature recommended by referee #1. In addition, the southeast wind continuously transports high $O_3$ from coastal to inland areas (from east to west) during this period (Figure 7a-d). But the lake breeze (from west to east) can hinder this process since they are in opposite direction (Figure R2g and h). The differences in deposition and transport processes finally lead to higher $O_3$ on the lake.

The high concentration of $O_3$ on the lake is like a reservoir because the onshore flow (from lake to land) of the lake breeze can transport the $O_3$ to the lakeside cities (Figure 9e and h). This is also the reason why $O_3$ concentration in lakeside cities will increase after the lake breeze is established.

[Figure]

**Figure R2.** Horizontal distributions of (a)-(c) gas-phase chemical reactions, (d)-(f) vertical mixing and (g-i) advection transport process to O₃ concentration in MODIS_noAH.

Reference

Park, R. J., Hong, S. K., Kwon, H. A., Kim, S., Guenther, A., Woo, J. H., and Loughner, C. P.: An
    evaluation of ozone dry deposition simulations in East Asia, Atmospheric Chemistry and
    Physics, 14, 7929-7940, 10.5194/acp-14-7929-2014, 2014.

4. Section 3.4.2. Only the sea breeze was discussed in this section. However, both the offshore and
onshore flows should be considered when we discuss circulations.

Response: Thanks for the constructive comment. During the daytime, the offshore flow (from land
to sea) is usually not obvious due to the strong background southeast wind (Figure 9). Therefore,

we don't mention the offshore flow in Sect. 3.3.2. However, the expansion of coastal cities, like Shanghai, can enhance the offshore flow. In the Sect. 3.4.2, as shown in Figure 11c, the offshore flow transports high concentration of $O_3$ to the sea, which may be an important reason for the forming of $O_3$-rich reservoir in the nocturnal residual layer on the sea. Thanks for your suggestion. These new contents have been added to the revised manuscript. Please see line 454-457.

5. Wind arrows in Figure 11 and 14 are too small to identify. Please improve the figure presentations with better quality.

Response: Thanks for the constructive comment. We take your suggestion, and enhance the length of the wind arrows in revised figures. Please see line 464 for new Figure 11 and line 525 for new Figure 13.

In fact, we almost redraw all the figures in the manuscript and the supporting information, hoping that they can meet the standard.

**Technical corrections**

1. "MODIS_withAH" and "MODIS_AH" should be the same, please choose any one of them.

Response: Thanks for your reminder. In the revised manuscript, we uniformly adopt "MODIS_AH" throughout the paper.

2. Line 258, "is" -> "are".

Response: Thanks for the constructive comment. The word "is" in the original manuscript is revised to "are". Please see line 254 in our new manuscript.

3. Line 277, "provide" -> "provides".

Response: Thanks for the constructive comment. The word "provide" in the original manuscript is revised to "provides". Please see line 277 in our new manuscript.

4. A few typos, grammatical and syntactic mistakes need to be corrected.

Response: We appreciate the referee for the valuable and constructive comments on our manuscript. We have utilized an English proofreading service through our university, hoping the written English is satisfactory in this version.

---

## Author Response (AR2)

**A point to point response to the editor's comments**

On behalf of my co-authors, we would like to express our great appreciation for your constructive comments and great effort on our manuscript entitled "Land use and anthropogenic heat modulate ozone by meteorology: A perspective from the Yangtze River Delta region" (acp-2021-619). We have studied the constructive comments carefully and have revised our manuscript. Replies to comments are in blue, corrections in the manuscript are in red. The line number in reply refers to the unmarked manuscript, in which all revisions have been accepted.

**Comments to the author**:

Dear Authors,

The reviewer agrees that the paper has largely improved and that, although the model results would benefit from updated land cover, the simulation can be accepted as it is. However, following the comments of the reviewer, I ask the author to extend the discussion/outlook mentioning that an update model coverage (GLC_2015) could improve the model results as shown by De Meij and Vinuesa (https://doi.org/10.1016/j.atmosres.2014.03.004) and Guang et al (https://doi.org/10.3390/su8070628).

Response: Thank you for the constructive comment and recommended literatures. In the revised manuscript, we add the content that uses an updated land cover dataset can improve the model results. More specifically, the underestimation of 2-m temperature ($T_2$) decreases when MODIS land use dataset and anthropogenic heat (AH) are taken into account. Please see lines 284-287. The simulation result of relative humidity (RH) is the worst with USGS land use dataset. Please see lines 294-295. The overestimation of 10-m wind speed ($WS_{10}$) is somewhat neutralized to fit the observations using MODIS instead of USGS land use dataset. Please see lines 301-303. The reasons for the model differences were discussed in the papers of De Meij and Vinuesa, and Guang, so their papers are cited in the revised manuscript. In addition, we extend an outlook mentioning that models can benefit from updated model coverage, like the GLC dataset. Please see lines 308-313.

Thanks and best regards

Wishing you a joyous Christmas and a prosperous New Year!

Thank you and best wishes.